# What Language is This? Ask Your Tokenizer.

**Clara Meister** [1]  **Ahmetcan Yavuz** [2]  **Pietro Lesci** [3]  **Tiago Pimentel** [2]

## Abstract

Language Identification (LID) is an important component of many multilingual natural language processing pipelines, where it facilitates corpus curation, training data analysis, and cross-lingual evaluation of large language models. Despite near-perfect performance on high-resource languages, existing systems remain brittle in low-resource and closely related language settings. We introduce UniLID, a simple and efficient LID method based on the UnigramLM tokenization algorithm, leveraging its probabilistic framing, parameter estimation technique and inference strategy. In short, to predict a string's language label, we simply ask: under which language's unigram distribution is this string most likely? Our formulation is data- and compute-efficient, supports incremental addition of new languages without retraining existing models, and can naturally be integrated into existing language model tokenization pipelines. Empirical evaluations against widely used baselines, including fastText, GlotLID-M, and CLD3, show that UniLID achieves competitive performance on standard benchmarks, substantially improves sample efficiency in low-resource settings—reaching $\sim$70% accuracy with as few as five labeled samples per language—and delivers large gains on fine-grained dialect identification.

## 1. Introduction

Language Identification (LID) systems are an important component of today's natural language processing (NLP) landscape. They are particularly important for the training and evaluation of large multilingual language models, where they are used to: (i) construct labeled corpora with adequate coverage of the world's languages, (ii) analyze the composition of large-scale training data, and (iii) evaluate models'

cross-lingual performance. As a result, reliable and efficient LID systems have become a vital tool for multilingual language modeling pipelines.

Simple and efficient algorithms have worked well for LID (Cavnar & Trenkle, 2001; Joulin et al., 2017; Lui & Baldwin, 2012); many regard it as a "solved" task since performance on high- and medium-resource languages is often near perfect (McNamee, 2005; Jauhiainen et al., 2019). However, LID systems often still perform poorly on low-resource languages, closely related language pairs, or fine-grained dialectal distinctions (Caswell et al., 2020; Chifu et al., 2024; Goot, 2025, *inter alia*). Even state-of-the-art language models fail to consistently identify less common languages (Kargaran et al., 2023). Collectively, these findings suggest that LID remains far from solved.

In this work, we propose a simple and computationally efficient approach to LID that combines the generative modeling framework underlying the UnigramLM tokenization algorithm with the classical Bayes decision rule. Concretely, UnigramLM assumes that a string is generated as a sequence of latent subword units drawn independently from a unigram distribution. Rather than learning a single global distribution, we estimate a separate unigram distribution per language. Given these language-specific distributions, we approximate the likelihood of an input string in each language using the probability of its most likely subword segmentation under the corresponding unigram model—intuitively, this allows segmentation itself to be treated as a language-dependent latent variable. Finally, we apply Bayes' rule to the resulting language-conditional likelihoods to obtain a posterior distribution over languages, and we assign the language label with the highest posterior probability.

The proposed method offers several advantages over prior LID approaches. While related to earlier generative LID models (Cavnar & Trenkle, 2001; Dunning, 1994), it departs in a key respect by treating segmentation as a language-specific phenomenon rather than enforcing a fixed tokenization across languages. This design choice encourages token boundaries that align with meaningful morphological structure rather than arbitrary statistical artifacts, an approach that is both linguistically motivated and empirically supported in prior NLP work (Bostrom & Durrett, 2020; Klein & Tsarfaty, 2020). The per-language distribution formu-

---

[1]EPFL [2]ETH Zürich [3]University of Cambridge. Correspondence to: Clara Meister <clara.meister@epfl.ch>.

*Proceedings of the 43$^{rd}$ International Conference on Machine Learning*, Seoul, South Korea. PMLR 306, 2026. Copyright 2026 by the author(s).

lation further enables incremental extension to new languages or dialects without retraining existing models, and empirically, only a few samples per language are needed to learn language-specific parameters that achieve strong performance. Finally, inference is computationally efficient: using clever dynamic programming tricks, inference can be done in an amount of time comparable to the inference step of the classic UnigramLM tokenization algorithm.

We compare UniLID against widely used LID systems: fastText, GlotLID-M, and CLD3. Across a diverse set of benchmarks, UniLID achieves performance competitive with these baselines while requiring substantially fewer labeled examples. The method is particularly effective in regimes where LID remains challenging. In low-resource settings, UniLID attains ∼70% accuracy with as few as five samples per language and ∼90% accuracy with fewer than 50 samples. On fine-grained dialect identification (Chifu et al., 2024), UniLID improves macro F1 from 0.53 to 0.72 relative to a fastText baseline. From a practical perspective, UniLID exhibits substantially shorter training times than fastText, while achieving comparable inference throughput, despite using a non-optimized research implementation. Taken together, these results suggest that treating segmentation as language-specific, rather than as a fixed preprocessing step, enables more effective LID systems.[1]

## 2. Language Identification

Let $\mathbf{s} = \langle s_1, \ldots, s_T \rangle$ be a string, *i.e.*, a sequence of characters[2] from an alphabet $\Sigma$, and $\Lambda$ a set of language labels. Closed-set LID is the task of assigning a language $\ell \in \Lambda$ to $\mathbf{s}$. This problem is often framed probabilistically, *i.e.*,

$$f_{\texttt{lid}}(\mathbf{s}) = \operatorname*{argmax}_{\ell \in \Lambda} \ p(\ell \mid \mathbf{s}) \tag{1}$$

and the task then becomes approximating $p$ with a learned model $p_\theta$. Two modeling paradigms are typically employed for solving this task. Discriminative approaches model this distribution using a score $f_\theta(\mathbf{s}, \ell) \in \mathbb{R}$, which is normalized using a softmax function:

$$p_\theta(\ell \mid \mathbf{s}) = \frac{\exp\{f_\theta(\mathbf{s}, \ell)\}}{\sum_{\ell' \in \Lambda} \exp\{f_\theta(\mathbf{s}, \ell')\}} \tag{2}$$

Generative approaches model this distribution via language-conditional likelihoods $p_\theta(\mathbf{s} \mid \ell)$; the desired distribution can then be computed via the Bayes rule:[3]

$$p_\theta(\ell \mid \mathbf{s}) = \frac{p_\theta(\mathbf{s} \mid \ell)}{\sum_{\ell' \in \Lambda} p_\theta(\mathbf{s} \mid \ell')} \tag{3}$$

---

[1]We release code for UniLID at https://github.com/Ahmetcanyvz/UNILID.

[2]The term "characters" denotes the atomic symbols of the input alphabet, encompassing both Unicode code points and bytes.

[3]We assume a uniform prior over languages in $\Lambda$ and so drop the explicit $p(\ell)$ term to reduce clutter.

Prior approaches to LID proposed different methods for parameterizing these distributions, as we discuss next.

### 2.1. Prior Approaches

$n$**-gram Models.** Early work on LID established the effectiveness of using plain character-level $n$-gram statistics with the generative modeling approach of eq. 3 (Cavnar & Trenkle, 2001; Dunning, 1994). Explicitly, they estimated language-conditional probabilities with character-level $n$-gram language models—often augmented with standard backoff or smoothing techniques (*e.g.*, Kneser–Essen–Ney smoothing; Ney et al., 1995). These models set a long-standing baseline for robust LID, particularly on short or noisy texts (Vatanen et al., 2010; Baldwin & Lui, 2010).

**Discriminative Frameworks.** In supervised settings, discriminative frameworks often showed improvements over earlier generative approaches, particularly when sufficient training data was available and languages were well represented (Jauhiainen et al., 2019). The standard approach is to represent input strings as feature vectors, encoding statistics of character $n$-grams such as term-frequency or TF-IDF weights. Formally, let $\gamma(\mathbf{s}) \in \mathbb{R}^d$ denote such a feature representation; this representation is then scored using a linear discriminative model, instantiating the framework of eq. 2, with a class-specific linear function $f_\theta(\mathbf{s}, \ell) = \theta_\ell^\top \gamma(\mathbf{s})$, where $\theta_\ell$ are language-specific parameters learned by minimizing the empirical cross-entropy over a labeled training set. A canonical example of this approach is fastText (Joulin et al., 2017). This architecture represents an input string as a bag of character $n$-grams, mapping each to a dense vector. These vectors are averaged to form a single hidden representation, which is then passed to a linear classifier to predict the language label. Despite its architectural simplicity, this approach provides a strong accuracy–efficiency trade-off and has become a standard reference point in LID research. Indeed, recent systems such as OpenLID and GlotLID-M combine this architecture with large-scale, carefully curated training data to achieve state-of-the-art coverage and reliability (Burchell et al., 2023; Kargaran et al., 2023). Subsequently, ConLID (Foroutan et al., 2025) applies supervised contrastive learning to the same architecture, encouraging representations of examples from the same language to be close and different languages to be separated.

**Neural Approaches.** In more recent years, neural approaches have been applied to LID. Most are based on character- and byte-level neural sequence models, which arguably provide a more flexible alternative to bag-of-$n$-gram representations such as fastText. Character-level CNN and bidirectional RNN baselines learn feature representations directly from raw text, eliminating the need for explicit feature engineering. These systems have shown competitive perfor-

mance, including for streaming or short text LID (Zhang et al., 2015; Belinkov & Glass, 2016; Kocmi & Bojar, 2017). Google's Compact Language Detector v3 (CLD3) is based on a neural classifier: input strings are first transformed into normalized character $n$-gram frequency features and then a shallow feed-forward network uses these features to predict a posterior over $\Lambda$. While multilingual Transformer encoders (*e.g.*, mBERT, XLM-R) have shown high accuracy in LID with task-specific fine-tuning, the substantial computational overhead and memory footprint of these architectures often render them impractical for high-throughput or resource-constrained environments.

### 2.2. Open Problems

**Varietal and dialectal discrimination.** Closely related languages or dialects (*e.g.*, Bosnian, Croatian, and Serbian; Hindi and Urdu; or Maghrebi and Levantine Arabic) have highly similar statistical properties, with a strong overlap in subword-structure and orthography. As a result, even state-of-the-art models struggle with distinguishing between them. Further, there is typically insufficient labeled training data to be able to learn these more fine-grained distinctions (Gaman et al., 2020; Chifu et al., 2024).

**Low-resource performance.** As with many NLP datasets, there exists much more data for a core set of high-resource languages than for the long tail of low-resource ones. Further, the data that does exist for these low-resource languages is often noisy, *e.g.*, contaminated with HTML code or snippets of text from high-resource languages (Kreutzer et al., 2022). Parameter estimates for these languages are often poor and, consequently, so is LID performance (Caswell et al., 2020; Kargaran et al., 2023). Reweighting, data augmentation, and contrastive objectives have shown promising improvements but have not yet eliminated performance gaps across different resource levels.(Ren et al., 2022; Burchell et al., 2023; Foroutan et al., 2025)

**Domain shift and orthographic noise.** Models trained on curated text collections often generalize poorly to informal or domain-mismatched inputs such as social media or transliterated scripts (Goot, 2025; Ojo et al., 2025). Many systems are also not robust to perturbations in orthography (*e.g.*, diacritic omission in languages like French or Vietnamese), as these substantially alter character $n$-gram statistics, leading to pronounced drops in both accuracy and calibration (Caswell et al., 2020).

**This work.** UniLID directly targets the first two challenges. As we show in (§6), it substantially improves dialect discrimination on DSL-ML 2024 (macro F1 $0.53 \rightarrow 0.72$) and yields large sample-efficiency gains in the low-resource regime ($\sim$70% accuracy with five samples per language).

On out-of-domain inputs it gives partial improvements; on orthographic noise we find UniLID and the strongest baseline to be roughly tied (§6).

## 3. UnigramLM

Tokenization is the process of segmenting strings into sequences of subword units, *i.e.*, character spans referred to as *tokens* in this context. Formally, let $\Sigma$ be the alphabet of characters $s$, and $\mathcal{V}$ the vocabulary of tokens $v$. Then, tokenization is the mapping of sequences of characters $\mathbf{s} = \langle s_1, s_2, \ldots \rangle \in \Sigma^*$ to sequences of tokens $\mathbf{v} = \langle v_1, v_2, \ldots \rangle \in \mathcal{V}^*$, which we denote as $\tau \colon \Sigma^* \to \mathcal{V}^*$. This mapping is typically injective and thus reversible: the original string can be obtained from its tokenized form through a deterministic detokenization function $\bot \colon \mathcal{V}^* \to \Sigma^*$, *i.e.*, $\mathbf{s} = \bot(\tau(\mathbf{s}))$. As each token (typically) represents a character span, the definition of $\bot$ is often simple, reducing to concatenating the characters within each token together, *i.e.*: $\bot(\langle v_1, v_2, \ldots \rangle) \overset{\text{def}}{=} v_1 \circ v_2 \circ \cdots$. For example, $\bot(\langle \text{'he', 'llo'} \rangle) = \textit{"hello"}$. Notably, the detokenization function is not necessarily 1-to-1: different token sequences may yield the same string. Reusing the example above, we observe that $\bot(\langle \text{'hell', 'o'} \rangle)$ still detokenizes to *"hello"*. The non-injectivity of $\bot$ is an aspect of tokenization that we will later take advantage of in our LID approach. Going forward, we refer to $\mathbf{v}$ as a **segmentation** of $\mathbf{s}$ if $\bot(\mathbf{v}) = \mathbf{s}$ and define $\mathcal{T}_{\mathcal{V}}(\mathbf{s}) = \{\mathbf{v} : \bot(\mathbf{v}) = \mathbf{s}\}$ as the set of all valid segmentations of $\mathbf{s}$ under vocabulary $\mathcal{V}$.

Numerous tokenization algorithms exist. These algorithms define a mapping function $\tau$ and the method for learning the parameters of this mapping function from data. For example, the Byte-Pair Encoding (BPE; Sennrich et al., 2016) algorithm defines a $\tau$ parametrized by a list of merges, and a method to learn this list. In this work, we focus on the UnigramLM algorithm (Kudo, 2018), which will form the basis of our LID approach.

### 3.1. Generative Model

In broad strokes, UnigramLM casts tokenization as the uncovering of a latent segmentation: it assumes strings are simply the surface form of a latent sequence of tokens $\mathbf{v}$, where each token is drawn independently from a categorical (unigram) distribution.

Let $\phi \in \Delta^{|\mathcal{V}|-1}$ denote a probability distribution over a vocabulary $\mathcal{V}$, where $\phi[v]$ gives the probability of token $v$. By construction, $\phi[v] \geq 0$ and $\sum_{v \in \mathcal{V}} \phi[v] = 1$. Under the UnigramLM framework, a token sequence of fixed length $M$ is generated by independently sampling each token from this distribution, and the likelihood of a token sequence

$\mathbf{v} = \langle v_1, ..., v_M \rangle$ is the joint probability of those tokens:[4]

$$v_m \overset{\text{i.i.d.}}{\underset{\text{for } m=1,...,M}{\sim}} \text{Categorical}(\boldsymbol{\phi}) \implies p_{\boldsymbol{\phi}}(\mathbf{v}) \overset{\text{def}}{=} \prod_{m=1}^{|\mathbf{v}|} \boldsymbol{\phi}[v_m] \quad (4)$$

A string $\mathbf{s}$ is then deterministically generated by detokenization. Consequently, the probability of $\mathbf{s}$ is obtained by marginalizing over all valid segmentations:

$$p_{\boldsymbol{\phi}}(\mathbf{s}) = \sum_{\mathbf{v} \in \mathcal{T}_{\mathcal{V}}(\mathbf{s})} p_{\boldsymbol{\phi}}(\mathbf{v}) \quad (5)$$

which formalizes the data-generating process of a string $\mathbf{s}$ under UnigramLM's assumptions. The posterior distribution over segmentations then follows naturally:

$$p_{\boldsymbol{\phi}}(\mathbf{v} \mid \mathbf{s}) = \frac{p_{\boldsymbol{\phi}}(\mathbf{v})}{p_{\boldsymbol{\phi}}(\mathbf{s})} \quad \text{for } \mathbf{v} \in \mathcal{T}_{\mathcal{V}}(\mathbf{s}) \text{ and } 0 \text{ otherwise} \quad (6)$$

Crucially, both $\boldsymbol{\phi}$ and $\mathcal{V}$ are unknown; we only observe the strings resulting from the generative process. The UnigramLM algorithm estimates these parameters from text data by combining the Expectation–Maximization (EM) algorithm with an iterative vocabulary pruning procedure.

### 3.2. Learning Model Parameters

The aim of the procedure proposed by the UnigramLM algorithm is to find the values of $\boldsymbol{\phi}$ and $\mathcal{V}$ that maximize data likelihood. Under the UnigramLM assumptions about the generative process of strings, the "complete" data consists of $(\mathbf{s}, \mathbf{v})$ pairs, *i.e.*, strings and the sequence of tokens that produced them. Since we only observe strings, and not their underlying segmentations, we cannot directly optimize for *complete*-data log-likelihood; we must instead optimize for the *observed*-data log-likelihood, where the observed data is our text corpus $\mathcal{C} = \{\mathbf{s}_k\}_{k=1}^K$:

$$\mathcal{L}(\mathcal{C}; \boldsymbol{\phi}) \overset{\text{def}}{=} \sum_{k=1}^{K} \log \sum_{\mathbf{v} \in \mathcal{T}_{\mathcal{V}}(\mathbf{s}_k)} p_{\boldsymbol{\phi}}(\mathbf{v}) \quad (7)$$

The quantity in eq. 7 is difficult to optimize directly due to the log-sum structure. The EM algorithm establishes a relationship between this quantity and the expected value of the complete-data log-likelihood, which allows us to solve an easier problem in its stead. In short, the EM algorithm iteratively optimizes parameters with respect to the expected value of the complete data log-likelihood given our observed data $\mathcal{C}$ and the distributions induced by our current belief of

model parameter values. In the context of UnigramLM, this boils down to performing the following two steps iteratively until our estimates $\boldsymbol{\phi}^{(n)}$ converge:[5]

1. **E-Step.** Let $c_v(\mathbf{v}) \overset{\text{def}}{=} \sum_{m=1}^{|\mathbf{v}|} \mathbb{1}\{v_m = v\}$, *i.e.*, the number of times token $v \in \mathcal{V}$ appears in a segmentation $\mathbf{v}$. Given current parameters $\boldsymbol{\phi}^{(n)}$ and fixed $\mathcal{V}$, the E-step computes expected counts in our corpus as

$$\widehat{c}_v(\mathcal{C}; \boldsymbol{\phi}^{(n)}) \overset{\text{def}}{=} \sum_{k=1}^{K} \sum_{\mathbf{v} \in \mathcal{T}_{\mathcal{V}}(\mathbf{s}_k)} p_{\boldsymbol{\phi}^{(n)}}(\mathbf{v} \mid \mathbf{s}_k) \, c_v(\mathbf{v}) \quad (8)$$

This computation is performed using dynamic programming, specifically via the forward-backward algorithm.

2. **M-Step.** The M-step then computes the new parameter estimates as simply the normalized expected counts of each token:

$$\boldsymbol{\phi}^{(n+1)}[v] = \frac{\widehat{c}_v(\mathcal{C}; \boldsymbol{\phi}^{(n)})}{\sum_{v' \in \mathcal{V}} \widehat{c}_{v'}(\mathcal{C}; \boldsymbol{\phi}^{(n)})}. \quad (9)$$

For a proof of why this is a valid approximation of MLE parameters (for a fixed $\mathcal{V}$), we refer the reader to the exposition on UnigramLM in Meister (2025).

Intuitively, this procedure resembles maximum likelihood estimation for a categorical distribution, but with a key difference: we cannot compute observed ("hard") counts of token occurrences, since the segmentation of each string into tokens is latent. We instead compute expected ("soft") token counts under the current model by marginalizing over all possible segmentations. In this way, the EM procedure integrates out segmentation uncertainty when estimating $\boldsymbol{\phi}$, yielding parameter estimates that reflect the full distribution over tokenizations rather than a single, fixed segmentation.

**Learning $\mathcal{V}$.** The EM procedure above operates over a fixed vocabulary $\mathcal{V}$. Indeed, the validity of the approximation relies on the fact that $\mathcal{V}$ is fixed (and known). In the case of our language identification algorithm, we can pre-specify it as, *e.g.*, the vocabulary of a chosen language model. If one wishes to also learn the vocabulary, the UnigramLM algorithm proposes a heuristic approach which works as follows. Initially, $\mathcal{V}$ is defined as an oversized set consisting of the most frequently occurring substrings in the corpus; this set can be found efficiently using Enhanced Suffix Array-based algorithms (*e.g.*, Abouelhoda et al., 2002). The vocabulary is then "learned" by iterating between: (i) holding it fixed and estimating $\boldsymbol{\phi}^{(n)}$ via EM; (ii) a pruning step, where the tokens whose removal hurts corpus log-likelihood the least—under current parameter beliefs $\boldsymbol{\phi}^{(n)}$—are removed. Via this pruning step, the algorithm ultimately trims the vocabulary down to the user's desired size.

---

[4]We would need either an EOS token (absorbing state) or a length prior in order to make this a valid probability distribution over $\mathcal{V}^*$. This detail is ignored in the standard UnigramLM implementation, and so we disregard it here as well for faithfulness to the original algorithm.

[5]$\boldsymbol{\phi}^{(0)}$ can be initialized as *e.g.*, the uniform distribution. See Land & Pinter (2025) for a discussion of this design choice.

### 3.3. Tokenizing Text

At inference time (*i.e.*, when turning text into tokens), the UnigramLM tokenization algorithm finds the most likely segmentation of $\mathbf{s}$ under the learned parameters $\widehat{\phi} = \phi^{(N)}$ as:[6]

$$\tau_{\widehat{\phi}}(\mathbf{s}) = \underset{\mathbf{v} \in \mathcal{V}^*}{\arg\max}\, p_{\widehat{\phi}}(\mathbf{v} \mid \mathbf{s}) \qquad (10\text{a})$$

$$\overset{\text{(eq. 6)}}{=} \underset{\mathbf{v} \in \mathcal{T}_{\mathcal{V}}(\mathbf{s})}{\arg\max}\, p_{\widehat{\phi}}(\mathbf{v}) \qquad (10\text{b})$$

$$\overset{\text{(eq. 4)}}{=} \underset{\mathbf{v} \in \mathcal{T}_{\mathcal{V}}(\mathbf{s})}{\arg\max}\, \prod_{m=1}^{|\mathbf{v}|} \widehat{\phi}[v_m] \qquad (10\text{c})$$

where we dropped $p_{\widehat{\phi}}(\mathbf{s})$ in eq. 10b since it does not depend on $\mathbf{v}$. The exact solution to eq. 10c can be found efficiently using a Viterbi-style algorithm, which runs in $\mathcal{O}(N \cdot T_{\max})$ time, where $N$ is the length of the input string $\mathbf{s}$ and $T_{\max}$ is the length of the longest token $v \in \mathcal{V}$.

## 4. UniLID

We propose a generative modeling approach for the LID task using the UnigramLM framework. In short, we define language-conditional distributions $p_\theta^\ell$ by learning unigram distribution parameters for each language $\ell$, albeit with a shared vocabulary across languages. We then compute language label probabilities, as in eq. 3, applying the Bayes' decision rule to ultimately assign the language label.

**Language-Conditional Distribution Estimation.** Let $\mathcal{V}$ be the vocabulary of a base tokenizer.[7] We fix $\mathcal{V}$ as our shared-across-languages vocabulary, but assume there exists, for each language $\ell$, a distinct unigram distribution $\widehat{\phi}_\ell$ over $\mathcal{V}$. Given a corpus $\mathcal{C}^\ell$ consisting of strings from language $\ell$, we then learn each of these distributions' parameters $\widehat{\phi}_\ell$ using the EM procedure described in §3.2. In other terms, similarly to as in standard UnigramLM, we compute an approximate maximum likelihood estimate by maximizing the observed-data log-likelihood, albeit on only a subset of the data:

$$\widehat{\phi}_\ell \overset{\text{def}}{\approx} \underset{\phi}{\arg\max}\ \mathcal{L}(\mathcal{C}^\ell; \phi) \qquad (11)$$

As in the standard UnigramLM formulation, this procedure integrates over segmentation uncertainty when estimating parameters. Because data from each language is processed separately, each language gets its own token frequency profile— and consequently its own set of optimal segmentations— despite sharing a common vocabulary across languages. We

---

[6]In the original UnigramLM algorithm, the most-probable segmentation is chosen rather than the one that has the highest marginal probability. We compared both strategies at inference and found no statistically significant accuracy difference, but marginalizing took roughly twice as much time (see §B.1).

[7]This does not need to be a UnigramLM tokenizer. Any tokenizer that uses a fixed vocabulary can be used.

discuss several design choices we make in our practical implementation of UniLID (*e.g.*, initialization of $\widehat{\phi}_\ell$ for the EM loop and floor values for token probabilities) in §5.

**Inference.** At inference time, we use the inference procedure of UnigramLM discussed in §3.3 to find the most probable segmentation of a string under each language's $\widehat{\phi}_\ell$. Importantly, this means that the most probable segmentation $\tau_{\widehat{\phi}_\ell}(\mathbf{s})$ can vary across languages. This yields a flexible model where the probability of a string reflects language-specific token frequencies, despite sharing a common vocabulary. We then approximate the probability of a string under language $\ell$ as the probability of its most probable segmentation in that language: $p_\theta(\mathbf{s} \mid \ell) = p_{\widehat{\phi}_\ell}(\tau_{\widehat{\phi}_\ell}(\mathbf{s}))$. Finally, we assign the language label that optimizes the posterior probability, *i.e.*, eq. 1. We write the instantiation of eq. 1 here explicitly for the reader's convenience:

$$f_{\texttt{lid}}(\mathbf{s}) = \underset{\ell \in \Lambda}{\arg\max}\, p(\ell \mid \mathbf{s}) = \underset{\ell \in \Lambda}{\arg\max}\, p_{\widehat{\phi}_\ell}(\tau_{\widehat{\phi}_\ell}(\mathbf{s}))$$

**Computational Complexity.** At training time, UniLID estimates a separate unigram distribution for each language by running the EM procedure over the corresponding language-specific corpus. For a corpus $\mathcal{C}^\ell$ of total character length $N_\ell$ (*e.g.*, number of Unicode code points or bytes), each EM iteration requires computing expected token counts via the forward–backward algorithm, which runs in $\mathcal{O}(N_\ell \cdot T_{\max})$, where again $T_{\max}$ denotes the maximum token length. Parameter updates are linear in the vocabulary size $|\mathcal{V}|$. Since per-language distribution parameters are estimated independently, training parallelizes trivially across languages. At inference time, UniLID uses a similar Viterbi-style dynamic program as in UnigramLM: first building a lattice that enumerates all valid segmentations of a string $\mathcal{T}_{\mathcal{V}}(\mathbf{s})$ and then finding the maximum probability option among them. The lattice needs only to be constructed once per string, regardless of the number of language-conditional distributions. Finding the optimal segmentation of a string and its probability for each language then takes $\mathcal{O}(|\Lambda| \cdot E)$ time, where $E$ is the number of edges in the lattice. Finally, we need a $\mathcal{O}(|\Lambda|)$ pass to evaluate the Bayes decision rule. In practice, this additional cost is modest relative to other standard components of modern NLP pipelines, *e.g.*, a single forward pass through a neural encoder, which typically dominates end-to-end runtime. Further, the number of edges $E$ in a lattice will be $N \cdot b$, where $b$ is a branching factor of the lattice, which tells us on average how many tokens in the vocabulary start at a certain string position and are part of one of its valid segmentations; typically this is on the order of 1-5 tokens, making the optimal-segmentation step much cheaper than lattice construction. In the pathological case, a lattice could have $b = T_{\max}$. In practice, $b \ll T_{\max}$ and so inference takes $\mathcal{O}(|\Lambda| \cdot N \cdot b + N \cdot T_{\max})$.

For reference, fastText and CLD3 share a similar input-processing cost linear in $N$: fastText extracts character n-grams over a sliding window ($\mathcal{O}(N \cdot k)$ for window size $k$). It then averages n-gram embeddings ($\mathcal{O}(N \cdot d)$) and scores all languages with a single matrix multiplication ($\mathcal{O}(d \cdot |\Lambda|)$); CLD3's scoring is asymptotically equivalent. The structural difference is that the language-scoring step in fastText and CLD3 is independent of input length, whereas UniLID's scales with $N$. In practice this is offset by the fact that the segmentation lattice is constructed once and reused across languages. We report empirical latency in §6.

## 5. Experimental Setup

We evaluate the performance of UniLID across several prominent LID benchmarks. We note that the majority of our training protocol and evaluation paradigms follow prior work (Kargaran et al., 2023; Foroutan et al., 2025). Our experiments compare against widely-used baselines: fastText, CLD3, and GlotLID-M. We perform analyses to show data efficiency, robustness to input length, and the impact of using different tokenizer vocabularies.

### 5.1. Datasets

We choose benchmarks that cover a spectrum of LID regimes, ranging from broad linguistic coverage and low-resource settings to fine-grained dialect identification: GlotLID-C (Kargaran et al., 2023) for large-scale coverage of the long tail of nearly 2,000 language–script labels; UDHR (Vatanen et al., 2010) and FLORES-200 (Costa-jussà et al., 2022) for content-controlled cross-lingual evaluation via parallel text across hundreds of languages; DSL-ML 2024 (Chifu et al., 2024) for fine-grained dialect identification across 14 regional varieties with high lexical overlap; Tatoeba (Tiedemann, 2020) for robustness on short, noisy, community-contributed text; and WiLI-2018 (Thoma, 2018) for controlled ablations, as its balanced 500/500 train/test split across 235 languages supports controlled comparisons across different dataset stratifications at manageable scale. We use official train/test splits where available; full per-dataset descriptions, including sample counts and split construction, are deferred to §A.

### 5.2. Baselines

We compare UniLID against the following systems. For a fair comparison, we hold the training dataset constant across systems that we train ourselves. For systems that did not release reproduction code, or for which we could not reproduce the associated paper's reported performance, we use the released checkpoints.

**fastText.** We train supervised fastText (Joulin et al., 2017) classifiers using the author's Python package[8] and following their recommended hyperparameter settings. Models were trained for 100 epochs.[9]

**CLD3 (pre-trained).** Google's Compact Language Detector v3 (CLD3) is a lightweight neural network that supports 107 languages and is optimized for short text segments and low-latency inference. We use the publicly available pre-trained model from their Python package.[10] Its language coverage does not fully overlap with that of our test sets. Thus, for a fairer comparison, we provide systems' results computed over only the subset of languages covered by CLD3 (right side of Table 1).

**GlotLID-M (pre-trained).** GlotLID-M is the model provided with the GlotLID-C benchmark dataset. Architecturally, it utilizes the fastText framework but differentiates itself through its training data curation. We use their openly available checkpoint, hosted on the same platform as the dataset. The exact training splits for GlotLID-M were not released but the authors state that UDHR and FLORES-200 are not part of the training set,[11] so we restrict GlotLID-M evaluations to these two benchmarks.

### 5.3. Evaluation Metrics

We report standard evaluation metrics for language identification, namely the macro-averaged F1 score and the macro-averaged false positive rate (FPR). Macro-averaging ensures that each language or dialect contributes equally to the overall score, regardless of class frequency. For analyses of data efficiency and robustness on the WiLI dataset, we additionally report classification accuracy.

### 5.4. UniLID Training

For each benchmark, we learn language-conditional unigram distributions using the EM procedure of the UnigramLM tokenization algorithm, described in §3.2. In all experiments, we initialize the unigram distribution as the uniform distribution and run EM for 20 rounds.[12] To avoid zero-probability segmentations when a token's expected counts

---

[8]https://pypi.org/project/fasttext.

[9]While the authors' base setup uses fewer epochs, models trained for fewer epochs seriously underperformed on benchmarks. We tuned fastText extensively. 100 epochs gave the best validation performance in every setting; see §C for results on our fastText hyperparameter sweep on DSL-ML 2024.

[10]https://github.com/google/cld3.

[11]Their contamination check finds that for 57 languages, $\leq 10\%$ of UDHR sentences appear in training; see §3 of Kargaran et al. (2023).

[12]We observed empirically that unigram distributions always converged within 20 rounds (*i.e.*, the total variation distance between subsequent rounds fell below $10^{-6}$).

*Table 1.* Classification results on benchmark evaluation sets. UniLID variants and fastText are trained using the GlotLID-C corpus. The left columns show results on the full test sets of each benchmark for models that have full coverage of those benchmark's language label set. The right columns show results on the subsets of the benchmarks that CLD3 has label coverage for.

| | Full GlotLID-C label set | | | | | | CLD3 label set | | | | | |
|---|---|---|---|---|---|---|---|---|---|---|---|---|
| **Method** | **GlotLID-C test** (1940 labels) | | **UDHR** (366 labels) | | **FLORES-200** (190 labels) | | **GlotLID-C test** (83 languages) | | **UDHR** (80 languages) | | **FLORES-200** (77 languages) | |
| | F1↑ | FPR↓ | F1↑ | FPR↓ | F1↑ | FPR↓ | F1↑ | FPR↓ | F1↑ | FPR↓ | F1↑ | FPR↓ |
| CLD3 | – | – | – | – | – | – | .906 | 5.48e-4 | .965 | 2.52e-4 | .978 | 2.22e-4 |
| GlotLID-M | – | – | **.871** | **1.37e-4** | **.968** | **1.35e-4** | – | – | .994 | 2.09e-5 | **.999** | **6.25e-6** |
| fastText | **.944** | 2.71e-5 | .855 | 1.87e-4 | .938 | 2.38e-4 | **.990** | 7.92e-5 | **.998** | 2.98e-5 | .998 | 2.96e-5 |
| UniLID | .929 | 2.03e-5 | .859 | 1.43e-4 | .932 | 2.78e-4 | .971 | 1.63e-4 | .992 | 1.06e-5 | .997 | 3.29e-5 |
| UniLID-Mistral-Nemo | .912 | **1.84e-5** | .836 | 1.79e-4 | .923 | 3.22e-4 | .972 | 1.53e-4 | .992 | **1.03e-4** | .994 | 8.05e-5 |
| UniLID-DeepSeek3.2 | .909 | 2.08e-5 | .836 | 1.78e-4 | .914 | 3.42e-4 | .971 | 1.66e-4 | .990 | 1.33e-4 | .994 | 7.92e-5 |
| UniLID-Qwen3 | .904 | 2.55e-5 | .829 | 1.91e-4 | .904 | 3.80e-4 | .964 | 2.27e-4 | .984 | 2.11e-4 | .989 | 1.46e-4 |

are estimated to be 0 under a given language (most common in the few-sample regime), we floor token probabilities at $1e - 12$. We did not find further smoothing measures to help. We consider two variants of UniLID: one that uses a dedicated base tokenizer trained on the LID training set, and one that uses the vocabulary of a pretrained language model; more details below.

**UniLID.** The base version of UniLID trains a tokenizer on the training split of each benchmark dataset. We train a byte-level tokenizer using the standard UnigramLM tokenization algorithm. Our implementation closely follows the design choices of the SentencePiece library (Kudo & Richardson, 2018), which provides the original implementation of UnigramLM. Unless otherwise specified, we use a vocabulary size of 100k; ablation experiments indicate that performance is relatively insensitive to this choice (see §B).

**UniLID-X.** This variant uses the vocabulary of a pretrained base tokenizer and learns only the per-language unigram distributions. This setting evaluates UniLID 's ability to operate with pretrained, task-agnostic vocabularies. We report results using the Mistral-Nemo-Base-2407[13] tokenizer (Jiang et al., 2023), which we refer to as UniLID-Mistral-Nemo. Results obtained with other base tokenizers, which yield comparable performance, are reported in §B.

## 6. Results

**Base Results.** We first evaluate UniLID on standard LID benchmarks: GlotLID-C, UDHR, and FLORES-200. Although UniLID is not always the top-performing system in terms of F1, Table 1 shows that both UniLID and UniLID-Mistral-Nemo achieve competitive macro-averaged F1 scores while consistently maintaining low FPR across both large-scale settings and low-resource datasets. Notably, on the largest evaluation set (full GlotLID-C test), UniLID

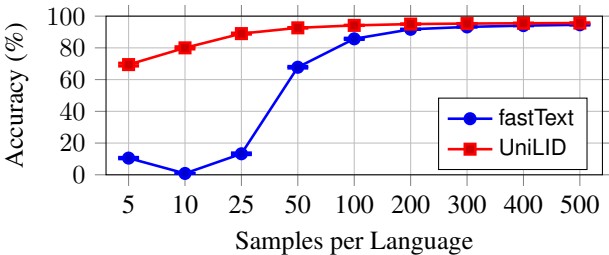

*Figure 1.* Sample efficiency analysis on WiLI (mean ± std). UniLID achieves substantially higher accuracy in the low-resource regime while maintaining consistently low variance across runs.

reduces FPR by roughly 25% compared to fastText (2.03e-5 vs 2.71e-5)—a property that can be more critical than F1 in many LID applications. For example, in web-scale crawling, a poor FPR for a low-resource language can lead to a training corpus dominated by noise (common languages misclassified as rare ones). Overall, these results demonstrate UniLID 's efficacy as a general-purpose LID system. In subsequent sections, we compare primarily against fastText. It is the only of these systems for which we can control training data (GlotLID-M is an instantiation of fastText, just with a particular dataset); we cannot control for test-set contamination or sample efficiency with the other systems.

**Dialect Differentiation.** Table 2 reports results on the DSL-ML 2024 benchmark. UniLID demonstrates strong performance in comparison to participant submissions in the DSL-ML 2024 shared task. On French dialects, UniLID achieves a macro-F1 of 0.534 in comparison to 0.385 for the top shared-task system. UniLID provides smaller but consistent gains on the other dialect groups as well: Spanish (0.850 vs 0.823), Portuguese (0.770 vs 0.752), and BCMS (0.769 vs 0.762). It also substantially outperforms fastText on macro F1, improving from 0.53 to 0.72. We note that part of this result is driven by the 0.00 F1 achieved by fastText on several of the Slavic languages. This poor performance may in part be due to the small amount of training data for those

---

[13]https://huggingface.co/mistralai/Mistral-Nemo-Base-2407.

*Table 2.* Per-dialect and macro-averaged F1 scores on the DSL-ML 2024 task for French (FR), Spanish (ES), Portuguese (PT), English (EN), and the Bosnian/Croatian/Montenegrin/Serbian family.

| | French | | | | Spanish | | Portuguese | | English | | BCMS | | | | |
|---|---|---|---|---|---|---|---|---|---|---|---|---|---|---|---|
| **Model** | BE | FR | CA | CH | ES | AR | BR | PT | US | GB | SR | HR | BS | ME | **Macro** |
| UniLID | .595 | .288 | .798 | **.456** | **.867** | **.832** | **.816** | **.724** | **.855** | **.815** | **.929** | **.788** | **.609** | **.750** | **.723** |
| fastText | **.697** | **.434** | **.811** | .444 | .833 | .686 | .729 | .609 | .815 | .561 | .833 | .000 | .000 | .000 | .532 |

*Table 3.* Macro-averaged F1 and false positive rate (FPR) of fastText and UniLID trained on WiLI and evaluated on the Tatoeba and UDHR benchmarks.

| Method | Tatoeba (201 langs) | | UDHR (142 langs) | |
|---|---|---|---|---|
| | **Macro F1**↑ | **Macro FPR**↓ | **Macro F1**↑ | **Macro FPR**↓ |
| fastText | 0.160 | 3.58e-3 | 0.849 | 6.07e-4 |
| UniLID | **0.414** | **9.61e-4** | **0.868** | **5.88e-4** |

*Table 4.* Accuracy as a function of input sequence length for UniLID and fastText on WiLI. All samples in WiLI are > 100 chars long.

| Length (chars) | Samples | UniLID Acc. (%)↑ | fastText Acc. (%)↑ |
|---|---|---|---|
| 101–150 | 7,845 | **93.10** | 90.73 |
| 151–200 | 26,652 | **94.17** | 92.56 |
| 201–300 | 31,449 | **95.86** | 94.58 |
| 301–500 | 29,494 | **96.78** | 96.03 |
| 501–1000 | 18,142 | **96.53** | 96.25 |
| 1000+ | 3,918 | **96.53** | 96.30 |
| **Overall** | 117,500 | **95.65** | 94.54 |

languages (see §A for the dataset breakdown), a regime that Fig. 1 likewise evinces that fastText struggles with. There are several potential reasons why UniLID performs particularly well in this regime: First, UniLID's vocabulary $\mathcal{V}$ provides a structural inductive bias; only per-language token frequencies must be estimated from the data. This stands in contrast to setups like fastText, where high-dimensional embeddings for millions of $n$-gram spans must be adjusted to give signal for the language. Second, generative classifiers are known to be more sample-efficient than discriminative ones in low-data regimes (Ng & Jordan, 2001), and UniLID is generative while fastText is discriminative.

**Low-resource Regime.** We evaluate sample efficiency on WiLI by varying the training dataset size. Concretely, we train each model on a subset of the WiLI training set, created by taking $K$ points from each language set. *E.g.*, at $K = 5$, there are a total of $5 \times |\Lambda|$ data points in the entire training set; training subsets are the same for each model at a given value of $K$. Fig. 1 shows system accuracy as a function of $K$. Results show that UniLID dramatically outperforms fastText[14] in the low-resource regime. With

---

[14]We tuned fastText extensively for this setting but could not improve performance; the best configuration is reported here, with a subset of our sweep of fastText hyperparameters in §C.

as few as 5 samples per language, UniLID achieves ∼70% accuracy, while fastText fails to generalize. Performance remains strong even with fewer than 50 samples per language, highlighting the strength of UniLID in scenarios in which labeled data is scarce or expensive to obtain. We provide the values displayed in Fig. 1 in Table 6 in §B.

**Out-of-domain Performance.** Table 3 evaluates generalization by training on Wikipedia (WiLI) and testing on short, noisy community-contributed text (Tatoeba) and formal legal documents (UDHR). UniLID demonstrates superior domain robustness, *more than doubling* the Macro F1 of fastText on Tatoeba (0.414 vs. 0.160). These results evince the generalization abilities of the UniLID approach, particularly for short, informal inputs, which is a domain in which current SOTA systems struggle.

**Robustness Analysis.** Table 4 reports accuracy as a function of input length on WiLI. UniLID consistently outperforms fastText across all length buckets, with the largest gains observed for shorter inputs, which is a regime in which most LID systems struggle. This highlights the robustness of UniLID's language-specific segmentation assumptions for low-context settings, which are common in practical LID applications, such as determining the language of social media posts. We additionally evaluate robustness to character-level corruption on WiLI by stochastically replacing non-whitespace characters at rates of 5% and 10%. Full setup details and results can be found in §B.2. In short, UniLID and fastText still perform well in the presence of light noise and degrade at very similar rates. While UniLID shows slightly better performance in the $< 10\%$ noise setting, neither system is meaningfully more robust than the other.

**Vocabulary Sensitivity.** Table 1 includes three UniLID-X variants that reuse off-the-shelf LLM tokenizer vocabularies (Mistral-Nemo, DeepSeek-3.2, Qwen-3); all are within 2.5 F1 of base UniLID on GlotLID-C, with similar behavior for other LLM tokenizers (§B). The gap in F1 is not negligible though. Clearly, training the base UniLID tokenizer on broad multilingual data, such that it attains a comprehensive multilingual vocabulary, does lead to better performance—both in-distribution (e.g., GlotLID-C train → GlotLID-C test) and out-of-distribution (GlotLID-C train → UDHR, FLORES-200, neither of which appears in the training set; see Table 1). LLM tokenizers, on the other

hand, are optimized for general-purpose language modeling on data distributions that under-represent the low-resource tail of GlotLID-C. We accordingly recommend, for production deployment, training the base tokenizer once on a broad, web-representative multilingual corpus, and then estimating per-language unigram distributions incrementally as data for new languages arrives. UniLID-X is intended as a convenience mode for practitioners who wish to reuse an existing LLM tokenizer or to integrate LID into an existing LLM tokenization pipeline at no additional vocabulary cost. We additionally explore the effect of vocabulary size in §B. Performance improves modestly with increasing vocabulary size before plateauing, suggesting that UniLID is relatively robust to this choice and does not require extremely large vocabularies to perform well.

**Efficiency.** §A reports inference latency and wall-clock training time for UniLID, fastText, and CLD3 measured on identical hardware. Training UniLID on the full GlotLID-C corpus (1,940 language–script classes) takes 17.8k seconds end-to-end versus ∼163k seconds for the 100-epoch fast-Text configuration, an approximately $9\times$ reduction despite our unoptimized research implementation. Inference latency varies with the label set size: on GlotLID-C (1,940 labels), UniLID is $1.65\times$ slower than fastText (0.274 vs 0.166 ms/sample) and $1.46\times$ slower than CLD3 (0.187 ms/sample); on WiLI (235 labels), UniLID is the second-fastest of the three (0.158 ms/sample, vs 0.113 for fastText and 0.427 for CLD3). The inference time dependency on label-set size can easily be mitigated in practice: for deployments with restricted target label sets, the generative formulation allows us to remove excluded languages trivially without changing the results. We discuss this next.

## 7. Discussion

**Limitations.** A main limitation of UniLID is its reliance on a unigram assumption, which ignores dependencies between neighboring tokens. Natural language is inherently contextual, and modeling token interactions is a critical part of most NLP approaches. In this respect, UniLID trades expressive power for computational and statistical efficiency. However, this design choice is conceptually aligned with widely adopted baselines such as fastText, which treats character $n$-grams as an orderless bag of features and thus likewise ignores neighboring token dependencies. The strong empirical performance of both methods suggests that a large fraction of the signal required for practical LID is captured by local subword statistics rather than long-range dependencies. Further, while UniLID is efficient at inference, its storage and memory requirements scale linearly with the number of modeled languages. This could prove potentially challenging for scenarios involving thousands of languages under strict latency constraints.

Notably, the modularity of our formulation means: (i) the per-language step parallelizes trivially across language partitions; (ii) the generative formulation allows the label set to be subsetted to a deployment's actual target languages without any retraining. Using these characteristics can drastically mitigate any inference and memory bottlenecks.

**Extensions and Future Work.** UniLID can be extended to text classification tasks outside of LID. Indeed, nothing in the formulation is specific to this task: estimating a separate unigram distribution per class over a shared vocabulary and scoring by Viterbi-segmentation applies to any partition of text with distinct subword statistics, such as domain, register, or source identification. Several other natural extensions and alterations to UniLID follow directly from the aforementioned limitations. First, the unigram assumption of UniLID could be relaxed so as to incorporate token dependencies. For example, one could define language-conditional $n$-gram models over token sequences—either baking this directly into the generative model formulation or learning $n$-gram models over texts segmented with the unigram generative model formulation. To mitigate the scaling of inference costs and memory with the size of the language set, several modifications could be explored: (i) hierarchical decoding: first selecting a script or language family, then discriminating within it would reduce the effective $|\Lambda|$. This could be done with something as simple as a script detection filter; (ii) sparse per-language distribution representations: in practice the per-language distributions are sparse after EM convergence for any given language (most directly, tokens in scripts a language does not use) so sparse representations could reduce the deployed footprint substantially.

## 8. Conclusion

We introduced UniLID, a simple and efficient approach for language identification. In short, to predict a string's language label, we ask: under which language's unigram distribution is this string most likely? More concretely, we adapt the generative framework of the UnigramLM tokenization algorithm, creating language-conditional probability distributions that enable segmentation to be treated as a language-specific phenomenon. A simple application of Bayes' rule to per-language string likelihoods then provides a posterior over language labels. Empirically, UniLID demonstrates performance competitive with widely used LID systems. In particular, it exhibits stronger data efficiency and greater robustness in low-resource and fine-grained identification regimes, suggesting it will be a valuable asset for curating high-quality, diverse datasets for next-generation multilingual language models.

## Impact Statement

This work aims to advance language identification in low-resource and fine-grained dialectal settings. As LID is often an initial step in multilingual data curation pipelines, the performance of these systems can strongly influence which languages and language varieties are supported by large-scale language models. Consequently, a potential positive impact of this work is to support more inclusive and representative multilingual datasets. By improving performance in low-resource and short-text regimes, the proposed method may facilitate the inclusion of languages and language varieties that are underrepresented in current large-scale corpora. The low false positive rate of our method is of particular importance, as data contamination in these settings can have a severe negative impact. Thus, we believe this method can contribute towards efforts to reduce the disparity in language model performance on high-resource vs. low-resource languages.

As with many language technologies, better LID capabilities raise dual-use concerns. While more accurate identification can facilitate inclusion, it can also be repurposed to filter, restrict, or suppress content associated with specific linguistic communities. In addition, fine-grained dialect identification could be misused for profiling or monitoring individuals or groups based on linguistic characteristics. These risks are not unique to the proposed method but are common to LID and related text classification technologies more broadly. We encourage practitioners to apply LID systems in ways that are consistent with ethical data collection and responsible use of language technologies.

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

# A. Additional Dataset Details

**GlotLID-C.** GlotLID-C (Kargaran et al., 2023) is a massive open-source benchmark designed to address the long-tail of LID. It covers 1940 different language–script classes,[15] significantly expanding the label space compared to traditional benchmarks. The dataset places a strong emphasis on low-resource and closely related language varieties. Train/test splits were not released by the dataset authors; we create our own splits following the pipeline of Foroutan et al. (2025): for each language, we split the data into 85%/15% for training/testing. We then subset the training set for each language to be at most $100k$ samples. Finally, we check explicitly for contamination and remove all occurrences of data from our eval only sets (described next) that we find in the training set. The resulting training set contains approximately $60M$ samples and test set contains approximately $45M$ samples of 1940 language–script combinations.

**UDHR (eval only; Vatanen et al., 2010).** The Universal Declaration of Human Rights (UDHR)[16] dataset contains parallel text across hundreds of languages. Because the semantic content is identical across all samples, UDHR is often used as a control dataset in LID; the parallel texts allow us to control for content difficulty when assessing per language performance. In LID, UDHR is only used as a test set, and its domain differs from most LID training sets, making it well-suited for testing generalization robustness.

**FLORES-200 (eval only; Costa-jussà et al., 2022).** FLORES-200 is another benchmark consisting of parallel texts. It covers 204 languages. Unlike crowdsourced datasets, FLORES-200 consists of professional translations of sentences extracted from Wikimedia projects (WikiNews, WikiVoyage, and WikiBooks). It thus provides a robust measuring stick against which we can assess low-resource language performance. We do not train on the FLORES-200 dataset; we evaluate on the `devtest` portion.

**DSL-ML 2024 (Chifu et al., 2024).** To evaluate performance on closely related languages, we use the DSL-ML 2024 shared task dataset. This benchmark focuses on fine-grained dialect identification with 14 labels covering regional variants of five major languages (English, French, Portuguese, Spanish, and South Slavic). It explicitly evaluates a domain in which lexical overlap is high and differentiating factors between dialects are nuanced. We use the train and dev splits released by the shared task, training on

---

[15]This is the number of unique labels in the officially maintained corpus on HuggingFace, which is what we employed: datasets/cis-lmu/glotlid-corpus.

[16]https://www.un.org/en/about-us/universal-declaration-of-human-rights

the train set and evaluating on the dev set.

| Dialect | Train Samples | Test Size |
|---|---|---|
| FR-BE | 122,369 | 7,543 |
| FR-CH | 116,490 | 1,027 |
| FR-FR | 84,402 | 6,351 |
| FR-CA | 19,041 | 2,169 |
| ES-ES | 2,616 | 631 |
| ES-AR | 1,982 | 358 |
| PT-PT | 1,331 | 356 |
| PT-BR | 2,556 | 635 |
| EN-US | 1,342 | 372 |
| EN-GB | 1,028 | 227 |
| SR | 236 | 86 |
| HR | 53 | 16 |
| BS | 45 | 16 |
| ME | 34 | 4 |

*Table 5.* Train and test set sizes by dialect on DSL-ML 2024. We provide these details to help explain the results observed in Table 2.

**WiLI-2018 (Thoma, 2018).** The Wikipedia Language Identification benchmark contains 235,000 paragraphs balanced equally across 235 languages, all derived from Wikipedia. We use WiLI-2018 to study data efficiency, base tokenizer choice, and the impact of input sequence length on classification accuracy, as it provides sufficient depth (1,000 samples per language, 500 train/500 test) to support statistically significant ablation studies. We use the official train/test splits.

**Tatoeba (eval only) (Tiedemann, 2020).** Tatoeba is a large, community-curated multilingual corpus of short, user-contributed sentences covering hundreds of languages and language varieties. The dataset is characterized by wide linguistic diversity, informal style, and substantial variation in sentence length and orthography, making it well suited for evaluating LID systems under realistic, noisy conditions. We use Tatoeba to assess robustness in low-resource and cross-lingual settings, where training data are limited and closely related languages frequently co-occur. As we use this dataset for testing out-of-domain robustness, we do not train on any portion of it. Models are evaluated on the entire dataset for the subset of languages that are in their label set. For evaluations with models trained on WiLI, this results in an evaluation on approximately $12M$ samples across 201 languages.

# B. Additional Ablation Results

*Table 6.* Accuracy (mean ± std) as a function of the number of training samples per language on WiLI.

| Samples / Language | UniLID Acc. (%) ↑ | fastText Acc. (%) ↑ |
|---|---|---|
| 5 | **69.46 ± 0.90** | 10.53 ± 0.43 |
| 10 | **80.01 ± 0.68** | 0.85 ± 0.00 |
| 25 | **88.99 ± 0.34** | 13.25 ± 0.53 |
| 50 | **92.62 ± 0.14** | 67.79 ± 0.52 |
| 100 | **94.17 ± 0.06** | 85.70 ± 0.40 |
| 200 | **94.99 ± 0.02** | 91.70 ± 0.04 |
| 300 | **95.31 ± 0.03** | 93.20 ± 0.04 |
| 400 | **95.50 ± 0.02** | 94.03 ± 0.03 |
| 500 | **95.65 ± 0.00** | 94.55 ± 0.06 |

*Table 7.* Effect of base tokenizer vocabulary size on UniLID performance and inference efficiency evaluated on WiLI.

| Vocab Size | Macro F1↑ | Macro FPR↓ | Latency (ms)↓ | Samples/s↑ |
|---|---|---|---|---|
| 10k | 0.945 | 2.514e-4 | 0.113 | 8891.99 |
| 20k | 0.951 | 2.278e-4 | 0.119 | 8421.68 |
| 50k | 0.957 | 2.019e-4 | 0.147 | 6818.08 |
| 100k | 0.960 | 1.859e-4 | 0.175 | 5717.68 |
| 200k | **0.9606** | **1.8382e-4** | 0.2328 | 4296.45 |

| Method | Macro F1↑ | Macro FPR↓ |
|---|---|---|
| UniLID (base) | **0.960** | **1.859e-4** |
| fastText | 0.946 | 2.331e-4 |
| UniLID-Mistral-Nemo | 0.958 | 1.925e-4 |
| UniLID-Mistral | 0.921 | 3.365e-4 |
| UniLID-LLaMA3.2 | 0.954 | 2.084e-4 |
| UniLID-LLaMA2 | 0.911 | 3.698e-4 |
| UniLID-DeepSeek3.2 | 0.955 | 2.042e-4 |
| UniLID-Qwen3 | 0.949 | 2.310e-4 |

*Table 8.* Macro-averaged F1 and FPR of different LID systems on WiLI; the latter 4 systems are variants of UniLID trained using the base tokenizers from open-source LLMs.

*Table 9.* Comparison of Viterbi decoding versus exact marginalization (forward algorithm) at inference time on GlotLID-C. Marginalization is approximately $2\times$ more expensive but yields no statistically significant gain.

| Decoding | Accuracy↑ | Macro F1↑ |
|---|---|---|
| UniLID (Viterbi) | 0.961 | 0.929 |
| UniLID (Marginalization) | **0.962** | **0.931** |

## B.1. Viterbi vs. Marginalization at Inference

UniLID uses Viterbi decoding at inference (§3.3), scoring each language by its most likely segmentation. An alternative is to compute the exact marginal likelihood $p(s \mid \ell)$ by summing over all segmentations using the forward algorithm. Table 9 compares the two strategies on GlotLID-C. The marginalized variant yields a negligible improvement

(within noise) at approximately $2\times$ the inference cost. Given the empirical near-tie, we retain Viterbi as the default inference method.

## B.2. Robustness to Orthographic Noise

To evaluate robustness under realistic input corruption, we apply stochastic character-level perturbations to the WiLI test set: with probability $p$, each non-whitespace character is independently replaced with another character drawn uniformly from the inventory of characters observed in the WiLI training set. We evaluate at $p \in \{0\%, 5\%, 10\%, 25\%, 50\%\}$. Example inputs at $p = 5\%$ and $p = 10\%$ are shown below; Table 10 reports accuracy, macro F1, and macro FPR for UniLID and fastText on the perturbed test sets.

**Example perturbations.**

**Original** *Anton (or Antonius) Maria Schyrleus (also Schyrl, Schyrle) of Rheita (1604–1660) was an astronomer and optician. He developed several inverting and erecting eyepieces. . .*

$p = 5\%$ *Anton (or Antonius) Maria Schyrlezs . . . Antonín . . . astronmmer and optijian. . .*

$p = 10\%$ *Anton (or Aston,us) MDria SchyrKeus . . . of iheith. . .*

At low noise ($p < 10\%$), UniLID maintains a small edge in F1 and accuracy. At moderate-to-high noise ($p \geq 25\%$), fastText degrades more gracefully than UniLID: the character n-gram representations underlying fastText appear to absorb localized character corruptions more robustly than UniLID's segmentation-based scoring, where corrupted characters can fragment otherwise-high-probability tokens. This points to a possible avenue for future work — explicit noise-aware token scoring or character-level smoothing within the UniLID framework.

# C. fastText Hyperparameter Sensitivity on Dialect Identification

To validate the fastText configuration used in our dialect experiments (§6), we ran a hyperparameter sweep over the number of training epochs and the minimum word-count threshold ($MC$) on the DSL-ML 2024 train/dev splits. Table 11 reports per-dialect F1 along with the macro-averaged F1 across all 14 labels. Performance improves monotonically with more epochs, with the 100-epoch + min-count=1000 configuration achieving the best macro F1. We use this best-performing configuration as our fastText baseline throughout the paper. Crucially, the three South Slavic dialects (HR, BS, ME) remain at 0.000 F1 across

*Table 10.* Performance under stochastic character-level perturbation on WiLI (117,500 samples, 235 languages). Each non-whitespace character is replaced with probability $p$. UniLID maintains a small edge at low noise; fastText degrades more gracefully at higher noise rates.

| | UniLID | | | fastText | | |
|---|---|---|---|---|---|---|
| **Noise $p$** | **Accuracy↑** | **Macro F1↑** | **Macro FPR↓** | **Accuracy↑** | **Macro F1↑** | **Macro FPR↓** |
| 0% | **0.957** | **0.960** | **1.86e-4** | 0.954 | 0.954 | 1.98e-4 |
| 5% | **0.951** | **0.954** | **2.10e-4** | 0.949 | 0.950 | 2.17e-4 |
| 10% | 0.940 | **0.944** | 2.55e-4 | **0.943** | 0.943 | **2.44e-4** |
| 25% | 0.824 | 0.832 | 7.51e-4 | **0.906** | **0.906** | **4.01e-4** |
| 50% | 0.395 | 0.396 | 2.58e-3 | **0.675** | **0.678** | **1.39e-3** |

*Table 11.* fastText per-dialect F1 on DSL-ML 2024 across different training configurations. Macro is the unweighted mean F1 across all 14 dialects. All three BCMS dialects (HR, BS, ME) remain at 0.000 F1 regardless of training budget.

| Setting | FR-BE | FR-FR | FR-CA | FR-CH | ES-ES | ES-AR | PT-BR | PT-PT | EN-US | EN-GB | SR | HR | BS | ME | Macro |
|---|---|---|---|---|---|---|---|---|---|---|---|---|---|---|---|
| 1 Epoch | 0.6906 | 0.2325 | 0.0000 | 0.2989 | 0.0000 | 0.0000 | 0.0000 | 0.3936 | 0.0000 | 0.0000 | 0.0000 | 0.0000 | 0.0000 | 0.0000 | 0.1154 |
| 5 Epochs | 0.6894 | 0.3766 | 0.2786 | 0.3997 | 0.8133 | 0.0240 | 0.3812 | 0.5323 | 0.7410 | 0.0000 | 0.0000 | 0.0000 | 0.0000 | 0.0000 | 0.3026 |
| 10 Epochs | 0.6832 | 0.4180 | 0.7779 | 0.4676 | 0.8580 | 0.0000 | 0.4995 | 0.5793 | 0.7657 | 0.0000 | 0.0000 | 0.0000 | 0.0000 | 0.0000 | 0.3607 |
| 10 Epochs, $MC$=10 | 0.6819 | 0.4163 | 0.8055 | 0.4956 | 0.8292 | 0.0000 | 0.0243 | 0.5620 | 0.7725 | 0.0000 | 0.0000 | 0.0000 | 0.0000 | 0.0000 | 0.3277 |
| 100 Epochs, $MC$=10 | **0.6908** | **0.4434** | 0.8139 | 0.4457 | **0.8674** | 0.5426 | 0.6088 | 0.6208 | 0.8065 | 0.5949 | 0.8293 | 0.0000 | 0.0000 | 0.0000 | **0.5189** |
| 100 Epochs, $MC$=1000 | .697 | .434 | **.811** | .444 | .833 | **.686** | **.729** | .609 | .815 | .561 | .833 | .000 | .000 | .000 | .532 |

*every* fastText configuration in the sweep, indicating that fastText fails on these closely related dialects regardless of training budget.

## D. Inference Latency and Training Time

Tables 12 and 13 reports end-to-end inference latency and throughput for UniLID, fastText, and CLD3 measured on the same hardware. On GlotLID-C ($|\Lambda| = 1940$), UniLID is approximately $1.65\times$ slower than fastText per sample; on WiLI ($|\Lambda| = 235$), the gap closes to $1.4\times$ and UniLID is over $2.7\times$ faster than CLD3. Table 14 compares wall-clock training time for the GlotLID-C training set.

*Table 12.* Inference latency on GlotLID-C ($|\Lambda| = 1940$), evaluated over 45,627,279 samples.

| Method | Latency (ms/sample)↓ | Throughput (samples/s)↑ |
|---|---|---|
| fastText | **0.166** | **6019** |
| CLD3 | 0.187 | 5356 |
| UniLID | 0.307 | 3253 |

*Table 13.* Inference latency on WiLI ($|\Lambda| = 235$), evaluated over 117,500 samples.

| Method | Latency (ms/sample)↓ | Throughput (samples/s)↑ |
|---|---|---|
| fastText | **0.113** | **8889** |
| UniLID | 0.155 | 6435 |
| CLD3 | 0.427 | 2340 |

*Table 14.* Wall-clock training time on the GlotLID-C training set. UniLID splits the training cost across base tokenizer learning (1k samples per language) and per-language EM re-estimation; fastText is trained on the full training set for 100 epochs.

| Method | Total Time (s)↓ |
|---|---|
| UniLID | **17,776** |
| fastText (100 ep.) | $\sim$163,000 |

## E. Performance Breakdown

In order to understand its strengths and weaknesses, we analyze UniLID's performance along three axes: writing script, training-data quantity, and segmentation length under predicted vs. true language.

### E.1. Script

Table 15 partitions GlotLID-C test performance by Unicode script. UniLID performs almost identically to fastText on Latin-script languages (1,700 of the 1,940 labels; F1 0.940 vs. 0.946), but underperforms on every non-Latin script. The gaps are largest on Greek ($\Delta$ F1 = -0.248), Hebrew (-0.227), Devanagari (-0.121), and Bengali (-0.100). We attribute this to the base vocabulary ($\mathcal{V}$) of the tokenizer: $\mathcal{V}$ is itself learned by UnigramLM on the GlotLID-C training corpus, in which Latin-script data dominates. A Latin-biased $\mathcal{V}$ leaves languages in other scripts with fewer informative subword units, which in turn limits how much signal per-language unigram estimation can recover. Larger vocabularies help modestly (§B) but do not close the gap. The most direct solution would be to train $\mathcal{V}$ on a script-balanced or script-upsampled corpus so that non-Latin substrings re-

| Script | # Langs | UniLID | fastText | Δ |
|---|---|---|---|---|
| Latn | 1,700 | 0.940 | 0.946 | −0.006 |
| Cyrl | 70 | 0.877 | 0.970 | −0.093 |
| Arab | 38 | 0.691 | 0.747 | −0.056 |
| Deva | 32 | 0.811 | 0.932 | −0.121 |
| Beng | 6 | 0.885 | 0.985 | −0.100 |
| Grek | 4 | 0.677 | 0.925 | −0.248 |
| Hebr | 4 | 0.740 | 0.967 | −0.227 |
| Armn | 2 | 0.974 | 0.986 | −0.012 |
| Other | 82 | 0.937 | 0.973 | −0.036 |

*Table 15.* Macro-F1 on the GlotLID-C test set stratified by Unicode script (ISO 15924). $\Delta = F1(\text{UniLID}) - F1(\text{fastText})$. UniLID matches fastText on Latin-script languages but underperforms on every non-Latin script, with the largest gaps on Greek, Hebrew, and Devanagari. We attribute this to the training distribution of the base vocabulary $V$, which is dominated by Latin-script text in GlotLID-C.

ceive proportional coverage in $\mathcal{V}$. We leave this exploration to future work.

### E.2. Resource Tier

Table 16 stratifies results by per-language training-sample count. UniLID is slightly exceeds fastText for languages with 500+ training samples. The differences arise in the very low resource settings: UniLID lags slightly on languages with fewer than 500 training samples. Many of the low-resource languages in GlotLID-C are from non-Latin scripts. Because our experiments on low-resource languages of the same script (§6) provide evidence that UniLID performs *better* than fastText in this setting, we hypothesize that the aforementioned script-bias issue (a property of the vocabulary) is a confounder for these sets of experiments.

### E.3. Length Bias

We lastly look potential length biases. The unigram likelihood lacks an explicit length prior, so a language whose $\widehat{\phi}_\ell$ puts more mass on longer tokens accumulates fewer multiplicative factors than one favoring shorter tokens. We checked whether this manifests as a systematic bias in errors and find that on misclassifications, the Viterbi segmentation under the predicted language is on average 0.17 tokens shorter than under the true language, with the gap growing for longer inputs. Length-normalizing the per-language log-likelihood by segmentation length led to worse performace in preliminary experiments; we report results in Tables 17 and 18.

| Train samples / lang | # Langs | $N_{\text{test}}$ | UniLID | | fastText | |
|---|---|---|---|---|---|---|
| | | | F1 | FPR | F1 | FPR |
| $<500$ | 56 | 2523 | 0.871 | $7.2 \times 10^{-5}$ | 0.915 | $1.15 \times 10^{-4}$ |
| $500-1k$ | 40 | 5246 | 0.975 | $1.5 \times 10^{-5}$ | 0.964 | $1.9 \times 10^{-5}$ |
| $1k-12k$ | 458 | 555 416 | 0.990 | $8.0 \times 10^{-6}$ | 0.979 | $8.0 \times 10^{-6}$ |
| $12k-18k$ | 526 | 1 157 747 | 0.997 | $2.0 \times 10^{-6}$ | 0.986 | $1.0 \times 10^{-5}$ |
| $18k-35k$ | 398 | 1 131 306 | 0.992 | $7.0 \times 10^{-6}$ | 0.981 | $1.6 \times 10^{-5}$ |
| $35k+$ | 462 | 42 775 041 | 0.958 | $5.3 \times 10^{-5}$ | 0.942 | $9.1 \times 10^{-5}$ |

*Table 16.* Performance on the GlotLID-C test set stratified by per-language training-sample count. # Langs is the number of languages in the bucket; $N_{\text{test}}$ is the total number of test instances. UniLID matches or exceeds fastText on every bucket except $<500$, where the base-vocabulary script bias documented in Table 15 compounds with limited per-language data.

| Length (chars) | N | Mean $\Delta$ | Median $\Delta$ | % fewer | % same | % more |
|---|---|---|---|---|---|---|
| All misclassified | 1,789,423 | $-0.17$ | 0.00 | 24.94 | 61.08 | 13.99 |
| $<30$ | 515,094 | $-0.11$ | 0.00 | 17.69 | 74.62 | 7.69 |
| 30–75 | 771,812 | $-0.15$ | 0.00 | 24.28 | 62.76 | 12.96 |
| 75–150 | 392,549 | $-0.21$ | 0.00 | 32.10 | 47.71 | 20.19 |
| 150–300 | 102,497 | $-0.24$ | 0.00 | 36.80 | 34.85 | 28.34 |
| 300+ | 7,471 | $-2.71$ | $-1.00$ | 53.09 | 15.11 | 31.80 |

*Table 17.* Token-count length bias on UniLID misclassifications, by input length. For each misclassified sample, $\Delta = n_{\text{pred}} - n_{\text{true}}$ is the difference in Viterbi segmentation length (in tokens) between the predicted and the true language; negative values mean the predicted language uses fewer tokens. Computed over all 1,789,423 misclassifications in the full GlotLID test set (45.6M samples). The mean $\Delta$ is negative and grows in magnitude with input length, while the median is zero and $61\%$ of errors leave the token count unchanged: a small but systematic bias toward fewer-token languages, driven by a minority of samples. *% fewer/same/more* give the share of misclassifications where the predicted language uses fewer, equal, or more tokens than the true language.

| Length (chars) | N | Original | Raw rescore | Normalized |
|---|---|---|---|---|
| $<30$ | 27,328 | 0.792 | 0.792 | 0.566 |
| 30–75 | 177,256 | 0.951 | 0.951 | 0.842 |
| 75–150 | 195,267 | 0.978 | 0.978 | 0.925 |
| 150–300 | 87,096 | 0.987 | 0.987 | 0.966 |
| 300+ | 13,053 | 0.995 | 0.995 | 0.991 |
| Overall | 500,000 | 0.960 | 0.960 | 0.885 |

*Table 18.* Effect of length-normalizing the per-language log-likelihood on UniLID accuracy, by input length. Scores are normalized by dividing the summed token log-probabilities by the segmentation length (score$/n_{\text{tokens}}^{\alpha}$ with $\alpha = 1$). *Raw rescore* re-runs the unnormalized scorer ($\alpha = 0$) through the same code path and reproduces the original predictions exactly (100% agreement), validating the implementation. Evaluated on a $500k$-sample subset. Full normalization lowers overall accuracy from 0.960 to 0.885, with the largest degradation on short inputs ($<30$ chars: $0.792 \rightarrow 0.566$) and almost none on long inputs: the opposite of where the token-count bias in Table 17 is largest.

