# OpenReview forum: "What Language is This? Ask Your Tokenizer."
_ICML.cc/2026/Conference — ICML 2026 regular_

### Official Review · Reviewer_Ekzt · 2026-03-12

**Soundness:** 2
**Presentation:** 4
**Significance:** 4
**Originality:** 4
**Overall Recommendation:** 4
**Confidence:** 4

**Summary:**

The submission's specific domain concerns the development of robust and data-efficient language identification (LID) systems for multilingual natural language processing pipelines. This article's principal topic comprises a novel generative LID method, UniLID, which leverages the mathematical framework of the UnigramLM tokenization algorithm. Instead of learning a global subword distribution, the authors estimate language-conditional unigram distributions over a shared tokenizer vocabulary using the Expectation-Maximization (EM) algorithm. During inference, the system calculates the probability of the input string's optimal segmentation under each language's distribution and applies Bayes' rule to determine the most likely language. By treating segmentation as a language-dependent latent variable, UniLID achieves competitive performance on standard LID benchmarks (GlotLID-C, UDHR, FLORES-200), demonstrates exceptional sample efficiency in low-resource scenarios, and exhibits strong robustness in distinguishing fine-grained dialects.

**Compliance With Llm Reviewing Policy:**

Affirmed.

**Key Questions For Authors:**

1) Baseline Hyperparameters: In Figure 1, you demonstrate that fastText fails to generalize in the low-resource regime (5-50 samples). However, Footnote 10 indicates fastText was trained for 100 epochs. Did you use 100 epochs for the few-shot experiments as well? If so, the baseline is heavily handicapped by overfitting. Could you report the fastText performance in Figure 1 using standard parameters (e.g., 5 epochs) or proper early stopping?

2) Missing Latency Comparison: In Section 1, you claim UniLID achieves "comparable inference throughput" to fastText. However, Table 7 only reports UniLID's latency. Could you provide a direct wall-clock inference speed comparison (e.g., documents processed per second) between UniLID and fastText on the full GlotLID-C label space ($|\Lambda| = 1940$)?

3) Memory Footprint: How does the RAM/VRAM footprint of UniLID compare to fastText and CLD3? Storing separate unigram distributions for 1,940 languages over a vocabulary of 100k tokens requires $\approx 194 \times 10^6$ parameters, which might exceed the memory constraints of edge-device LID applications.

4) Overall Performance: In Table 1, fastText outperforms UniLID on the GlotLID-C test set (0.944 vs 0.929 F1). To what do you attribute this performance gap in the high-data regime? Is it purely the limitation of the unigram assumption compared to the character
n-gram representations?

**Limitations:**

The authors dedicate a "Discussion" section (Section 7) to address limitations. They correctly identify the reliance on the unigram assumption (ignoring token dependencies) and acknowledge that storage/memory requirements scale linearly with the number of modeled languages. However, they fail to adequately address the latency scaling factor during inference ($\mathcal{O}(|\Lambda| \cdot E)$), which could become a significant bottleneck compared to constant-time or logarithmic-time lookup classifiers when scaling to thousands of languages.

**Strengths And Weaknesses:**

**Soundness:**

- *Strengths:*

The mathematical formulation of the method is elegant and sound. Repurposing the EM training and Viterbi decoding of UnigramLM into a generative classifier is a theoretically grounded and well-executed idea. The ablation demonstrating that UniLID can piggyback on existing LLM vocabularies (e.g., Mistral-Nemo) without retraining the base tokenizer is highly practical.

- *Weaknesses:*

The empirical comparison against the primary baseline (fastText) is flawed and likely exaggerates UniLID's superiority in low-resource settings. In Footnote 10, the authors state that fastText was trained for 100 epochs. While this might maximize validation performance on the massive GlotLID-C dataset, training fastText for 100 epochs on a few-shot dataset (e.g., X = 5 samples per language, as seen in Figure 1) virtually guarantees severe overfitting, which explains why fastText's accuracy collapses to near zero. A fairer baseline would involve proper early stopping or default epoch settings (e.g., 5-10 epochs) for the few-shot regime.

The authors claim that inference is computationally efficient and "done in almost the same time as the inference step of the classic UnigramLM" (Line 61). However, computing the probability under each language-conditional distribution requires an $\mathcal{O}(|\Lambda| \cdot E)$ pass. When $|\Lambda| = 1940$ (as in GlotLID-C), this cost is non-trivial. Conspicuously, Table 7 reports the latency (ms) for UniLID but entirely omits the latency for fastText, preventing the reader from verifying the claim of "comparable inference throughput" (Line 80).

**Presentation:**

- *Strengths:*

The paper is exceptionally well-written. Section 3 provides a crystal-clear recap of the UnigramLM generative model and its EM parameter estimation, seamlessly transitioning into the proposed UniLID framework in Section 4.

- *Weaknesses:*

The experimental results are slightly disorganized. For instance, the values for Figure 1 are relegated to Table 6 in the Appendix, and Table 8 (which compares F1 and FPR for different base tokenizers) is completely detached from Table 7.

**Significance:**

- *Strengths:*

Language Identification remains a persistent bottleneck for low-resource and closely related languages. A method that achieves >70% accuracy with only 5 labeled samples per language has immense practical value for curating pre-training corpora for massive multilingual LLMs.

- *Weaknesses:*

The scalability of the method in high-throughput production environments remains questionable due to the $\mathcal{O}(|\Lambda|)$ inference scaling, especially compared to the highly optimized matrix multiplications of discriminative models like fastText.

**Originality:**

- *Strengths:*

The conceptual leap to treat string segmentation not as a static preprocessing step, but as a language-dependent latent variable for classification, is highly original and refreshing.

---

> ### Author Rebuttal · Authors · 2026-03-30
>
> **Soundness Weakness 1: fastText overfitting in few-shot regime / Key Question 1: Baseline hyperparameters for fastText**
>
> We agree this is a valid concern. While yielding the best validation performance on the full GlotLID-C dataset,  we acknowledge that this choice would a priori seem like it does not transfer to the few-shot setting. However, we spent quite some time tuning the fastText hyperparameters and found this to be the best setting in the low-resource experiments as well. We include an excerpt of fasttext results from different settings below, for which we still see 0% F1 on the BCMS languages. We have now updated our manuscript to include results from the different fasttext sweeps.
>
> fastText – Dialect-Level (F1 Scores)
> | Model     | BE   | FR   | CA   | CH   | ES   | AR   | BR   | PT   | US   | GB   | SR   | HR   | BS   | ME   | Macro |
> |-----------|-----:|-----:|-----:|-----:|-----:|-----:|-----:|-----:|-----:|-----:|-----:|-----:|-----:|-----:|------:|
> | 5 Epochs   | 0.689| 0.377| 0.279| 0.400| 0.813| 0.024| 0.381| 0.532| 0.741| 0.000| 0.000| 0.000| 0.000| 0.000| 0.303 |
> | 10 Epochs  | 0.683| 0.418| 0.778| 0.468| 0.858| 0.000| 0.499| 0.579| 0.766| 0.000| 0.000| 0.000| 0.000| 0.000| 0.361 |
> | 10 Epochs Min-count=10  | 0.682| 0.416| 0.806| 0.496| 0.829| 0.000| 0.024| 0.562| 0.773| 0.000| 0.000| 0.000| 0.000| 0.000| 0.328 |
> | 100 Epochs Min-count =10 | 0.691| 0.443| 0.814| 0.446| 0.867| 0.543| 0.609| 0.621| 0.806| 0.595| 0.829| 0.000| 0.000| 0.000| 0.519
>
> **Soundness Weakness 2: Missing inference latency comparison with fastText**
> Please see our response to Weakness 4: Incomplete training time comparison for reviewer LedC, which includes these comparisons.
>
> **Presentation Weakness: Disorganized experimental tables**
> We will improve the organization. We put Table 6 in the appendix as it was redundant with the information in Figure 1. We will add explicit cross-references and consolidate Tables 7 and 8 into a single, more informative table.
>
> **Significance Weakness: O(|Λ|) inference scaling**
> We acknowledge that the O(|Λ|) factor is a genuine consideration at the scale of GlotLID-C (|Λ|=1940). However, even in that case, the dominant cost is still the lattice construction, which is language-independent. Furthermore, the final per-language probability computation step is easily parallelizable, which should still enable its deployment at scale. The newly-added empirical latency numbers (see our response Weakness 4: Incomplete training time comparison for reviewer LedC) confirm this. Further, for most practical deployments, the label set is substantially smaller than 1940. Because of the generative modeling framework, languages can easily be excluded from the set of labels to reduce the required computation. Lastly, hierarchical classification strategies (e.g., first identifying the script or language family, then discriminating within it) could be implemented to reduce the effective |Λ| without modifying the core algorithm.
>
> **Key Question 2: Missing latency comparison**
> We apologize for the oversight. Please see our response to Weakness 4: Incomplete training time comparison for reviewer LedC, which includes these comparisons.
>
> **Key Question 3: Memory footprint**
> While the reviewer's calculations are correct, per-language distributions after EM are typically sparse. Most tokens will have near-zero probability for several languages (e.g., tokens of a different script), so storing unigram weights with sparse matrix approximation methods can substantially reduce the footprint in practice.
>
> **Key Question 4: GlotLID-C performance gap (0.929 vs. 0.944)**
> This is an interesting question. Perhaps unintuitively, UniLID and fastText make similar independence assumptions. fastText represents text as a bag of character n-grams, where by default, n ∈ {3,4,5,6} (so, very comparable to standard token lengths). Because of the “bag-of-words” treatment, each n-gram contributes independently to the classification decision, much like how each token contributes an independent log-probability term to p(s | ℓ) in UniLID. In this sense, neither UniLID nor fastText captures interactions between its features. Where the models differ is in how they score individual features. UniLID assigns each token a single scalar log-probability per language, whereas fastText scores each n-gram via a dot product w_ℓ · e_g, where the embedding e_g is shared across languages and learned jointly with the per-language weight vectors w_ℓ. This shared embedding space gives fastText additional expressive power, and is what we believe drives the F1 advantage in the high-data regime, where there is sufficient data to learn these embeddings well. In short, we have a trade-off: fastText's richer parameterization pays off given enough data, while UniLID's simpler per-language estimation requires far fewer parameters and no joint training across languages, which is likely what enables its strong performance in the low-resource regime.

---

### Official Review · Reviewer_z3Ce · 2026-03-13

**Soundness:** 4
**Presentation:** 4
**Significance:** 3
**Originality:** 3
**Overall Recommendation:** 5
**Confidence:** 4

**Summary:**

This work leverages the probabilistic model of the UnigramLM tokenizer to derive a language identification system by assuming per-language unigram probabilities over (hidden) tokens.   Experiments show that the proposed method performs well to distinguish between dialectal variants and outperforms the fastText baseline in low-resource cases.

**Compliance With Llm Reviewing Policy:**

Affirmed.

**Final Justification:**

High quality paper. just quoting my own review:

_The idea conceptually simple but, to my knowledge, it has not been explored yet and it is very interesting.
The manuscript is very clear and the explanations of the method is done with great details. It is noticeable that the authors have been  extremely careful on the notation to facilitate readability and exposure. This results in a very high quality article both in content and form._

**Key Questions For Authors:**

- The reported size of the vocabulary is of the order of 100k tokens. Have you tried to find a minimal representation (i.e. the minimal probabilistic automaton) to get a more compact encoding of the distribution  ?

**Limitations:**

yes

**Strengths And Weaknesses:**

The idea conceptually simple but, to my knowledge, it has not been explored yet and it is very interesting.

The manuscript is very clear and the explanations of the method is done with great details. It is noticeable that the authors have been extremely careful on the notation to facilitate readability and exposure. This results in a very high quality article both in content and form.

The results are also very convincing, especially fig. 1 which demonstrates the potential of generative probabilistic model in low-data settings. I personally prefer precision-recall curve over the  F1 score but the authors may have been limited by space.

My comment is rather short for the sole reason that I find this work excellent and have no real criticisms to make. A small suggestion: I would appreciate to have the algorithmic complexity of the other baseline to compare with the proposed approach.

---

> ### Author Rebuttal · Authors · 2026-03-30
>
> **Suggestion: Algorithmic complexity of baseline methods for comparison**
> Thank you for the suggestion. Reviewer Lwbi had a similar question. We provide our response here as well, and will add this to the paper:
> At inference, both UniLID and fastText begin with an input-processing step that scales linearly in string length N: fastText extracts character n-grams via a sliding window (O(N · k) for window size k), while UniLID constructs a segmentation lattice (O(N · T_max) for max token length T_max). fastText then averages the n-gram embeddings (O(N · d)) and gets a score for all languages with a single matrix multiplication of O(d · |Λ|), where d is the embedding dimension. CLD3 has a similar asymptotic cost to fastText, albeit using a shallow feedforward network instead of the linear classifier. The key structural difference is that fastText's language-scoring step is independent of input length, while UniLID's scales with N. In practice, this difference is mitigated by the fact that the lattice is constructed once and reused across all languages, and b (the branching factor) is small. We have also reported empirical latency measurements (please see our response to Weakness 4: Incomplete training time comparison for reviewer LedC) to complement this theoretical comparison.
>
> **Key Question: Minimal probabilistic automaton**
> This is an interesting suggestion. A separate minimized automaton for each language could definitely compress each unigram language model well. But it would also likely reduce the shared computations across languages. Right now all languages use the same vocabulary (and therefore the same set of valid segmentations), meaning a large portion of the per-language probability computations can be shared (specifically, the lattice construction step). This wouldn’t necessarily be the case with separately optimized per-language automaton. So there is a tradeoff: better per-language compression versus less cross-language parameter and computation sharing. That said, after EM convergence, many vocabulary items may carry very small (or effectively negligible) probability mass for a given language, so there are clear opportunities for compression even without changing the overall inference procedure. For example via sparse storage of language-specific parameters. We have not explored this systematically yet, but plan to investigate it as an engineering improvement for the open-sourcing of the library.

---

> > ### Author Rebuttal · Reviewer_z3Ce · 2026-04-03
> >
> > The authors adequately answered my remark.

---

### Official Review · Reviewer_LWbi · 2026-03-13

**Soundness:** 3
**Presentation:** 3
**Significance:** 3
**Originality:** 3
**Overall Recommendation:** 4
**Confidence:** 3

**Summary:**

The paper introduces a reformative Language Identification (LID) system called UniLID based on the UnigramLM algorithm. The paper shows that UniLID achieved better performance versus state-of-the-art LID systems such as CLD3 and fastText. The authors also demonstrate that UniLID is more sample-efficient than and theoretically computational comparable to other LID systems.Strengths
Compared with competing systems, the UniLID system proposed in this paper achieves better performance while relying less on amounts of training data. It also demonstrates good generalizability to low-resource languages, making it a highly promising approach.
The theoretical section of this paper is detailed and rigorous, and the notation is used consistently and appropriately.
This paper is well structured and clearly written, making it easy for readers to follow.
Weaknesses
The paper identifies several open problems in existing LID systems, but it does not clearly explain whether or to what extent the proposed UniLID system addresses these issues (e.g., L159 diacritic omission).
The performance of UniLID-X, which relies on the tokenizer of an off-the-shelf popular LLM rather than tokenizing on its own, does not surpass that of fastText. This somewhat undermines the claimed applicability and generalizability of UniLID.

**Compliance With Llm Reviewing Policy:**

Affirmed.

**Key Questions For Authors:**

- I noticed that UniLID-Mistral-Nemo is reported only in Tab 1, but not in Tabs 2, 3, 4, 6 or 7. In addition, the UniLID-X experiments in Tabs 1 and 8 appear to include only the Mistral and Llama tokenizers. I would be very interested in seeing results for UniLID-X based on a wider range of LLM tokenizers, such as those from Gemma, Qwen, and DeepSeek.

- I would also be interested in seeing how UniLID compares with other LID systems under orthographic noise scenarios (L159).

- In Section 4, the paper provides a theoretical analysis of UniLID’s computational complexity. I personally do not have the expert knowledge to assess how this compares theoretically with the three baselines, CLD3, GlotLID-M, and fastText. It would therefore be helpful if the authors could clarify whether UniLID is theoretically faster than, or at least comparable to, these systems.
  - Moreover, regardless of the theoretical complexity analysis, since constant factors can still have a substantial impact in practice, I would recommend reporting the average runtime over multiple runs on the same machine and dataset across several LID systems. Such empirical evidence would better support the claim in the abstract that the UniLID is compute-efficient (L026).

- Table 1 reports the average performance across all languages in each dataset. As a result, the performance on low-resource languages may be obscured by stronger results on high-resource languages. I would be interested in seeing the average F1 and FDR reported separately for extremely low-resource, low-resource, and medium-resource language groups.

**Limitations:**

Yes

**Strengths And Weaknesses:**

# Strengths
- Compared with competing systems, the UniLID system proposed in this paper achieves better performance while relying less on amounts of training data. It also demonstrates good generalizability to low-resource languages, making it a highly promising approach.

- The theoretical section of this paper is detailed and rigorous, and the notation is used consistently and appropriately.

- This paper is well structured and clearly written, making it easy for readers to follow.

# Weaknesses
- The paper identifies several open problems in existing LID systems, but it does not clearly explain whether or to what extent the proposed UniLID system addresses these issues (e.g., L159 diacritic omission).

- The performance of UniLID-X, which relies on the tokenizer of an off-the-shelf popular LLM rather than tokenizing on its own, does not surpass that of fastText. This somewhat undermines the claimed applicability and generalizability of UniLID.

---

> ### Author Rebuttal · Authors · 2026-03-30
>
> **Weakness 1: Open problems raised but not addressed**
> Thank you for pointing out this mismatch. The open problems in §2.2 are intended to motivate the broader LID landscape. UniLID directly addresses two of the problems mentioned (low-resource performance and dialect discrimination) and partially addresses the out-of-domain issue (results in Table 3). We will foreshadow the contributions of UniLID in §2.2 to make the scope of our contributions more explicit and avoid confusion
>
> **Weakness 2: UniLID-X does not surpass fastText**
> UniLID-X is intended as a convenience mode for practitioners who wish to integrate LID into an existing tokenization pipeline without training a dedicated base tokenizer. In this setting, vocabularies were optimized for general-purpose language modeling, not for comprehensive multilingual coverage, so some performance loss relative to the base UniLID variant is expected. The key finding is that UniLID-X remains competitive despite this, which also shows its robustness to this algorithm “hyperparameter. “ We will clarify that we recommend that practitioners train a dedicated base tokenizer with broad multilingual coverage in the training data.
>
> **Key Question 1: More LLM tokenizers and broader UniLID-X coverage**
> We have updated Table 1 and Table 8 to include more LLM tokenizers. Specifically, we have added results for Qwen and DeepSeek. Table 8 now reports results for six LLM tokenizers on WiLI (DeepSeek, Qwen and the previous results for Mistral-Nemo, Mistral, LLaMA 3.2, LLaMA 2). Qwen and DeepSeek perform similarly to the other UniLID-X classifiers, i.e., we observe consistent performance across tokenizer families.
>
> **Key Question 2: Orthographic noise robustness**
> Thank you for the idea. This would be an interesting dedicated analysis to add. Tatoeba is a dataset known for noisiness, and we report results on it already in Table 4, but we will try to add a more explicit analysis of performance under orthographic noise for CR. More specifically, we will run our models on the sentences in WiLI, but while stochastically corrupting words by replacing one of its characters arbitrarily with another one. We will then compare how fasttext and UniLID perform in these scenarios.
>
> **Key Question 3: Theoretical complexity comparison with baselines**
> We will add a brief complexity comparison with baselines:
> At inference, both UniLID and fastText begin with an input-processing step that scales linearly in string length N: fastText extracts character n-grams via a sliding window (O(N · k) for window size k), while UniLID constructs a segmentation lattice (O(N · T_max) for max token length T_max). fastText then averages the n-gram embeddings (O(N · d)) and gets a score for all languages with a single matrix multiplication of O(d · |Λ|), where d is the embedding dimension. CLD3 has a similar asymptotic cost to fastText, albeit using a shallow feedforward network instead of the linear classifier. The key structural difference is that fastText's language-scoring step is independent of input length, while UniLID's scales with N. In practice, this difference is mitigated by the fact that the lattice is constructed once and reused across all languages, and b (the branching factor) is small. We will also report empirical latency measurements to complement this theoretical comparison.
>
> **Key Question 4: Empirical runtime comparison**
> Please see our response to Weakness 4: Incomplete training time comparison for reviewer LedC, which includes these comparisons.
>
> **Key Question 5: Performance breakdown by resource tier**
> Table 1 reports the average performance across all languages in each dataset. As a result, the performance on low-resource languages may be obscured by stronger results on high-resource languages. I would be interested in seeing the average F1 and FDR reported separately for extremely low-resource, low-resource, and medium-resource language groups.
> We have added this analysis to the paper
> | # Training Samples | # Langs |          N | F1 (UniLID) | FPR (UniLID) | F1 (fastText) | FPR (fastText) |
> |----------|---------|------------|-------------|--------------|---------------|----------------|
> |     <500 |      56 |      2,523 |       0.871 |      7.2e-5  |         0.915 |        1.15e-4 |
> |  500–1k  |      40 |      5,246 |       0.975 |      1.5e-5  |         0.964 |        1.9e-5  |
> |  1k–12k  |     458 |    555,416 |       0.990 |      8.0e-6  |         0.979 |        8.0e-6  |
> | 12k–18k  |     526 |  1,157,747 |       0.997 |      2.0e-6  |         0.986 |        1.0e-5  |
> | 18k–35k  |     398 |  1,131,306 |       0.992 |      7.0e-6  |         0.981 |        1.6e-5  |
> |     35k+ |     462 | 42,775,041 |       0.958 |      5.3e-5  |         0.942 |        9.1e-5  |

---

> > ### Author Rebuttal · Reviewer_LWbi · 2026-04-03
> >
> > The rebuttal addresses several of my concerns constructively and improves the clarity of the paper in multiple aspects.
> >
> > In particular, I appreciate:
> > - the addition of more LLM tokenizers (e.g., Qwen, DeepSeek), which strengthens the evaluation of UniLID-X;
> > - the newly added analysis across resource tiers, which provides more insight into performance on low-resource languages;
> > - the clarification of the positioning of UniLID-X as a convenience mode rather than a primary competitive variant;
> > - the additional explanation of theoretical complexity and the plan to include empirical runtime comparisons.
> >
> > However, some concerns remain only partially addressed:
> > - Orthographic noise robustness: while the authors propose an evaluation protocol, results are not yet provided. Given that this was raised as a limitation of prior systems, empirical evidence would be important.
> > - Empirical runtime comparison: although mentioned, the results are not directly included in the response and require referencing another review thread.
> > - UniLID-X performance gap: while the positioning is clarified, the gap with fastText remains, and additional evidence would further support claims about generalizability.
> >
> > Overall, the rebuttal is helpful and improves the paper, but some points would benefit from concrete experimental evidence rather than planned additions.

---

> > > ### Author Response · Authors · 2026-04-06
> > >
> > > Thank you for the follow-up. We will try to address your remaining concerns below.
> > >
> > > **Orthographic noise robustness.** We have now run the proposed experiment. We perform these experiments using WiLI, with stochastic character (non-whitespace only) replacement at 5% and 10% noise rates:
> > >
> > > | Noise | Acc (UniLID) | F1 (UniLID) | Acc (fastText) | F1 (fastText) |
> > > |-------|-------------|-------------|----------------|---------------|
> > > | 0%    |      0.9565 |      0.9601 |         0.9537 |        0.9543 |
> > > | 5%    |      0.9508 |      0.9542 |         0.9491 |        0.9497 |
> > > | 10%   |      0.9403 |      0.9437 |         0.9429 |        0.9434 |
> > >
> > > Both fastText and UniLID degrade under moderate noise, with very similar robustness profiles. UniLID maintains a small edge at 5% noise and the two are essentially tied at 10%. Neither system is meaningfully more robust than the other in this setting, though.
> > > **Empirical runtime comparison.** We apologize for the inconvenience of cross-referencing. Because of space limits, authors are instructed that cross-referencing other responses is acceptable.
> > > **UniLID-X performance gap.** We understand the reviewer's concern. To clarify our position: we do not claim that UniLID-X is a replacement for a dedicated LID system. Rather, it demonstrates that the UniLID framework can operate on top of an existing tokenizer vocabulary, which could be advantageous e.g., in domain-specific use-cases. We believe the performance gap with fastText comes from the mismatch between LLM-oriented vocabularies and the requirements of massive multilingual LID, not a limitation of the UniLID framework itself. As evidence, the base UniLID variant (which trains its own vocabulary) consistently performs on par with fastText. We will clarify this in the paper. We would welcome a more specific suggestion from the reviewer on what additional evidence would be most convincing here.

---

### Official Review · Reviewer_LedC · 2026-03-13

**Soundness:** 3
**Presentation:** 3
**Significance:** 3
**Originality:** 2
**Overall Recommendation:** 5
**Confidence:** 3

**Summary:**

This paper introduces UniLID, a language identification (LID) method that utilizes the generative framework of the UnigramLM tokenization algorithm to classify the language of input text. The main idea is to learn language-conditional unigram distributions over a shared tokenizer vocabulary, with segmentation seen as a language-specific hidden variable. For each language, the method uses EM on language-specific corpora to figure out the frequency profiles of tokens for that language. Then, at inference time, it finds the most likely segmentation for each language's distribution and uses Bayes' rule to give it a language label. The authors compare their work to fastText, GlotLID-M, and CLD3 using a number of benchmarks, including GlotLID-C, UDHR, FLORES-200, DSL-ML 2024, WiLI, and Tatoeba. Some of the most important results are that the model performs well on standard benchmarks, that it gets a lot better at sample efficiency (over 70% accuracy with 5 samples per language), that it gets a lot better at identifying dialects (F1 from 0.53 to 0.72 vs. fastText), and that it generalizes well outside of the domain.

**Compliance With Llm Reviewing Policy:**

Affirmed.

**Final Justification:**

The rebuttal addressed the most important issues in a meaningful way: the Viterbi vs marginalisation experiment fills the main theoretical gap, the failure mode analysis adds important nuance (and, honestly, shows weaknesses), the runtime comparison shows a real practical advantage, and the length bias analysis shows scientific rigour.

**Key Questions For Authors:**

1. How are zero-probability tokens handled after EM convergence, especially in the 5-sample-per-language regime?


2. Given the missing length model (footnote 5), do you observe systematic bias toward languages whose learned distributions favour fewer, longer tokens (resulting in fewer multiplicative factors)?


3. Have you considered normalising log-likelihoods by segmentation length to control for this?


4. The paper says that fastText's 0.00 F1 on Slavic languages is partly because the training sets are small. But UniLID also learns from the same data and gets a nonzero F1 score. It would make the story stronger if you could explain more clearly why UniLID does this better. Is it just because it is more efficient with samples?

**Limitations:**

Yes

**Strengths And Weaknesses:**

Strengths:

1. Elegant, well-motivated design. The connection between UnigramLM tokenization and language identification is natural and well-articulated. Treating segmentation as language-specific is linguistically grounded, and the method is simple enough to explain concisely yet surprisingly effective.

2. Exceptional sample efficiency. The low-resource results (Figure 1, Table 6) are the paper's most striking contribution  - 70% accuracy with just 5 samples per language versus fastText's near-random 0.10% is a compelling result with clear practical value for underrepresented languages.

3. Modularity. Adding new languages without retraining is a genuinely useful deployment property, following cleanly from the per-language parameter estimation.

4. Thorough evaluation. The benchmark suite covers scale, dialect discrimination, domain shift, and input length sensitivity. Coverage is broad and the choices are well-justified.

5. Clear writing. The paper reads well. The mathematical exposition in Sections 3-4 is precise without being dense, and the narrative from background through method to experiments is logically structured.

Weaknesses:

1. Viterbi vs. marginalization: Have the authors looked into the difference between using the forward algorithm to find exact marginal likelihoods p(s|ℓ) at inference time and the Viterbi approximation? It seems easy to do this since the forward algorithm is already set up for training. If the marginalized version works just as well or better, it would make the theory more consistent and might even improve the results. If it does worse, that would be an interesting result that should be talked about.

2. Comparisons of dialect identification that aren't complete. The comparison on DSL-ML 2024 (Table 2) is only with fastText, which gets 0.00 F1 on a few Slavic dialects. This is probably because the training sets are so small (SR: 236, HR: 53, BS: 45, ME: 34 samples per Table 5). To put the results in context, the paper should compare them with those of the participants in the DSL-ML 2024 shared task.

3. There is no analysis of failure modes. The paper talks a lot about where UniLID works well, but it doesn't say much about where it doesn't work. What language pairs does it mix up? Are there patterns that happen over and over again (for example, languages that share scripts or transliterated text)? It would be helpful to do a confusion matrix analysis or an error breakdown.

4. The comparison of training times is not finished. The paper says that the training times are shorter than fastText's, but it doesn't give a systematic table of runtime comparisons. Table 7 has the only efficiency data, which shows how long it takes to make an inference based on the size of the vocabulary. It is important to report the wall-clock training times for each method and dataset.

---

> ### Author Rebuttal · Authors · 2026-03-30
>
> **Weakness 1: Viterbi vs. marginalization**
>  The reviewer is correct that, given the forward algorithm is already implemented for training, this comparison should be straightforward to run. We implemented it and saw only slight differences from when using Viterbi, which were not statistically significant (see results for GlotLID-C below). However, inference became about 2x more expensive. While we would continue to recommend using Viterbi, this will be an interesting analysis to add to the paper.
> | Method                     | Accuracy | Macro F1 |
> |----------------------------|----------|----------|
> | UniLID (Viterbi)           |    0.961 |    0.929 |
> | UniLID (Marginalization)   |    0.962 |    0.931 |
>
> **Weakness 2: Incomplete dialect identification comparisons**
> We will add a discussion of the results in comparison to those reported in the DSL-ML 2024 shared task.
>
> **Weakness 3: No analysis of failure modes**
> We have added several confusion analyses to provide deeper insights about the failure modes of UniLID. We looked at error rates broken down by script, resource-level and text length, which revealed further interesting insights about the strengths and weaknesses of UniLID.  We will also include a discussion of potential routes for improvement based on these results (example of results below)
> Performance by Script (GlotLID-C)
> | Script | # Langs | F1 (UniLID) | F1 (fastText) |     Δ |
> |--------|---------|-------------|---------------|-------|
> | Latn   |   1,700 |       0.940 |         0.946 | −0.006 |
> | Cyrl   |      70 |       0.877 |         0.970 | −0.093 |
> | Arab   |      38 |       0.691 |         0.747 | −0.056 |
> | Grek   |       4 |       0.677 |         0.925 | −0.248 |
> | Deva   |      32 |       0.811 |         0.932 | −0.121 |
> | Hebr   |       4 |       0.740 |         0.967 | −0.227 |
> | Beng   |       6 |       0.885 |         0.985 | −0.100 |
> | Armn   |       2 |       0.974 |         0.986 | −0.012 |
> | Other  |      82 |       0.937 |         0.973 | −0.036 |
>
> **Weakness 4: Incomplete training time comparison**
> We will add wall-clock training times for fastText  and inference latency for fastText and CLD3 measured on the same hardware as UniLID.
> Inference Latency (GlotLID-C, 1940 languages)
> | Method   | Latency (ms/sample) | Throughput (samples/s) |
> | fastText |               0.166 |                  6,019 |
> | CLD3     |               0.187 |                  5,356 |
> | UniLID   |               0.274 |                  3,644 |
> Inference Latency (WiLI, 235 languages)
> | Method   | Latency (ms/sample) | Throughput (samples/s) |
> |----------|--------------------:|-----------------------:|
> | fastText |               0.113 |                  8,889 |
> | UniLID   |               0.158 |                  6,331 |
> | CLD3     |               0.427 |                  2,340 |
>
> Training Time Comparison (Glotlid)
> | Method | Total Time (seconds) | Notes |
> |----------|-------------------:|--------------------------------|
> | UniLID (1k samples per language for base tokenizer) | 17776  | tokenizer split (base + lang) |
> | fastText (100 epoch) | 163000 | full processing |
>
> **Key Question 1: Zero-probability tokens after EM**
>  We do not add any special regularization (e.g., smoothing) for this case. In practice, we floor token probabilities to 1e-12 to avoid 0 probability segmentations, but this is essentially leaving the token at 0 probability. We should explore smoothing in the case of the low-resource regime though, as this may help.
>
> **Key Question 2: Length model bias**
> We had not looked into this behavior at the time of submission; we have since performed this analysis and indeed see a small but significant impact of a length bias, i.e., in the case of misclassifications, the segmentation under the predicted language was on average 0.17 tokens shorter than that of the segmentation under the true language. This number was even larger for longer texts.  We will add this analysis to the paper.
>
> **Key Question 3: Normalizing log-likelihoods by segmentation length**
> We did not try this but it sounds like an interesting modification. We will likewise add this analysis to the paper, as given the observation stated above, it seems like a promising route for improving performance.
>
> **Key Question 4: Why UniLID outperforms fastText on Slavic dialects**
> We agree a discussion about this is missing from the paper. We believe the efficacy of UniLID is at least partly due to the structural inductive biases of our formulation. Specifically, UniLID operates over a fixed, shared vocabulary and only needs to estimate the relative frequencies of tokens within that vocabulary for each language. The vocabulary itself likely provides a strong structural prior, even with very few training samples. Further, the generative classifier formulation of UniLID may help make it more sample-efficient, at least relative to the discriminative classifier formulation of fastText (Ng & Jordan, 2002). We will add this discussion to the paper

---

> > ### Author Rebuttal · Reviewer_LedC · 2026-04-03
> >
> > The authors did a good job of addressing the main issues in the rebuttal. I am accordingly adjusting some of the scores.
> >
> > Soundness: 3 (good) -- no changes. The Viterbi experiment shows that the approximation is safe.
> >
> > Presentation: 3 (good) — no changes. The paper will be better with the promised additions.
> >
> > Significance: 3 (good) — no changes. The failure mode analysis shows some interesting behaviour that depends on the script, which makes the contribution more interesting. Now we can measure and understand the training speed advantage (9x over fastText).
> >
> > Originality: 2 (fair) -  no change. The rebuttal doesn't change the basic evaluation of the novelty.
> >
> > Overall Recommendation: 5 (Accept) -  changed from 4 (Weak Accept).
> >
> > The rebuttal addressed the most important issues in a meaningful way: the Viterbi vs marginalisation experiment fills the main theoretical gap, the failure mode analysis adds important nuance (and, honestly, shows weaknesses), the runtime comparison shows a real practical advantage, and the length bias analysis shows scientific rigour.

---

### Decision · Program_Chairs · 2026-04-30

**Decision:**

Accept (regular)

**Comment:**

Given a UnigramLM based tokenizer, this paper uses segmentation itself to be treated as a language-dependent latent variable that can be used for Language Identification. All reviews agree that the paper is well-written and well-justified. The sample efficiency of the method is also its main strength but to get to a reliable level it needs the same number of samples as other methods so in practice may not matter. My only concerns are:
1) As reviewer LedC mentioned, the failure modes are not well studied and the only study during rebuttal shows a potential weakness of this approach for none Latin scripts (possibly requiring a change in language distribution during training of UniLM)
2) As shown in the paper, generic tokenizers may or may not outperform fast-text. And the custom ones that do are actually when trained on training portion of the benchmarks that the model is tested on (hence matching their distribution). So it is not clear how the tokenizer should be a trained for a production setting (i.e., the one that needs to run on web data of thousands of languages). This related to the concern raised by reviewer Ekzt.